# The supernova remnant SN 1006 as a Galactic particle accelerator

Roberta Giuffrida[1,2], Marco Miceli [1,2] ✉, Damiano Caprioli[3], Anne Decourchelle [4], Jacco Vink [5], Salvatore Orlando [2], Fabrizio Bocchino[2], Emanuele Greco[2,5,6] & Giovanni Peres [1,2]

The origin of cosmic rays is a pivotal open issue of high-energy astrophysics. Supernova remnants are strong candidates to be the Galactic factory of cosmic rays, their blast waves being powerful particle accelerators. However, supernova remnants can power the observed flux of cosmic rays only if they transfer a significant fraction of their kinetic energy to the accelerated particles, but conclusive evidence for such efficient acceleration is still lacking. In this scenario, the shock energy channeled to cosmic rays should induce a higher post-shock density than that predicted by standard shock conditions. Here we show this effect, and probe its dependence on the orientation of the ambient magnetic field, by analyzing deep X-ray observations of the Galactic remnant of SN 1006. By comparing our results with state-of-the-art models, we conclude that SN 1006 is an efficient source of cosmic rays and obtain an observational support for the quasi-parallel acceleration mechanism.

Cosmic rays (CRs) are extremely energetic particles, mainly composed by protons. It is widely accepted that the bulk of cosmic rays (below the knee at approximately $3 \times 10^{15}$ eV) stems within our Galaxy, playing a significant role in its energy budget[1]. Several convincing indications point towards supernova remnants (SNRs) as their most likely source[2].

Shock waves in SNRs can accelerate particles through the first-order Fermi mechanism (or diffusive shock acceleration, DSA). The capability of SNRs of accelerating electrons is testified by the ubiquitous synchrotron radio emission detected at their shock fronts[3], associated with approximately 1–10 GeV electrons. X-ray synchrotron emission from ultrarelativistic (about 10 TeV) electrons has been first detected in the remnant of the supernova observed in 1006 AD[4] (hereafter SN 1006) and, afterward, in other young SNRs[5]. Accelerated protons in SNRs might provide $\gamma -$ ray emission through interaction with protons in their environment leading to $\pi^0$ production and subsequent decay. TeV $\gamma -$ ray emission from a handful of shell-like SNRs has been observed[6], while GeV emission has been detected in about 30 remnants[7]. In SN 1006, $\gamma -$ ray emission has been detected in both the TeV and GeV bands, with the *HESS* observatory[8] and *Fermi* telescope[9], respectively, but its nature is uncertain, since $\gamma -$ ray emission can also be produced by inverse Compton from ultrarelativistic electrons (leptonic scenario). Indications for a hadronic origin of the $\gamma -$ rays have been obtained in a few cases[10–13], confirming that SNR shocks can indeed accelerate hadrons.

The identification of SNRs as the main Galactic factory of CRs is still based on plausibility arguments and many important open issues need to be addressed[14]. To prove that SNRs are CR factories, it is necessary to show that they indeed supply the power needed to sustain the Galactic CRs (of the order of $10^{41}$ erg s$^{-1}$). Considering the rate of supernovae in our Galaxy (about 2 per century), SNRs should transfer about 10–20% of their characteristic kinetic energy (approximately $10^{51}$ erg) into CRs[15]. The loss of such a large fraction of the ram energy is expected to alter the shock dynamics with respect to the adiabatic case. Nonlinear DSA predicts the formation of a shock precursor (travelling "ahead" of the main shock) that modifies the shock structure. This shock modification should result in an increase of the

[1]Università degli Studi di Palermo, Dipartimento di Fisica e Chimica E. Segrè, Piazza del Parlamento 1, 90134 Palermo, Italy. [2]INAF-Osservatorio Astronomico di Palermo, Piazza del Parlamento 1, 90134 Palermo, Italy. [3]Department of Astronomy and Astrophysics & Enrico Fermi Institute, The University of Chicago, 5640 S Ellis Ave, Chicago, IL 60637, USA. [4]Université Paris-Saclay, Université Paris Cité, CEA, CNRS, AIM, 91191 Gif-sur-Yvette, France. [5]Anton Pannekoek Institute, GRAPPA, University of Amsterdam, PO Box 94249, 1090 GE Amsterdam, The Netherlands. [6]GRAPPA, University of Amsterdam, Science Park 904, 1098 XH Amsterdam, The Netherlands. ✉e-mail: marco.miceli@unipa.it

total shock compression ratio, $r_t$, and a decrease of the post-shock temperature with respect to the Rankine-Hugoniot (adiabatic) values[16–19]. Recent self-consistent hybrid (kinetic ions and fluid electrons) simulations show that efficient acceleration of CRs leads also to the formation of a shock postcursor where non-linear magnetic fluctuations and CRs drift away from the shock front, moving downstream[20,21]. This postcursor is quantitatively more important for shock modification than the precursor; it acts as an additional energy sink, providing an increase of $r_t$, even with a moderate acceleration efficiency: when the CR pressure is about 5–10% of the bulk ram pressure, the total compression ratio ranges approximately between $r_t = 5$–7.

SN 1006 is an ideal target to reveal shock modification. Thanks to its height above the Galactic plane (approximately 600 pc), the remnant evolves in a fairly uniform environment (in terms of density and magnetic field). Moreover, SN 1006 shows regions with prominent particle acceleration, where shock modification might be present, together with regions where there are no indications of particle acceleration and where we do not expect shock modification. In particular, the bilateral radio, X-ray and $\gamma$ − ray morphology of SN 1006 clearly reveals nonthermal emission in the northeastern and southwestern limbs. X-ray synchrotron emission in nonthermal limbs, seen notably above 2.5 keV, identifies sites of electron acceleration to TeV energies, whereas the lack of nonthermal emission in the southeastern and northwestern limbs marks regions without considerable particle acceleration, where the X-ray emission is mainly thermal (see Fig. 1a). It has been shown that the efficiency of diffusive shock acceleration increases by decreasing the angle $\theta$ between the ambient magnetic field and the shock velocity[22]. There is strong evidence that the bilateral morphology of SN 1006 can be explained in the framework of this quasi-parallel scenario, with the ambient magnetic field, $\mathbf{B}_0$, oriented approximately in the southwest-northeast direction[23–25]. If these nonthermal limbs are also sites of efficient hadron acceleration, the signature of shock modification is expected to be stored in the X-ray emission of the shocked interstellar medium[17] (ISM). The amount of shock modification, namely an increase in the post-shock density, should raise near the nonthermal limbs (i.e., in quasi-parallel conditions), being smaller in thermal regions, where the shock velocity and $\mathbf{B}_0$ are almost perpendicular. Shock modification is expected to reshape the remnant structure, by reducing the distance between the forward shock and the outer border of the expanding ejecta (i. e., the "contact discontinuity"). However, this effect (observed in SN 1006[26,27]) can also be explained as a natural result of ejecta clumpiness[28], without the need of invoking shock modification.

A first attempt to observe azimuthal variations in shock compressibility was performed by analyzing the *XMM-Newton* X-ray

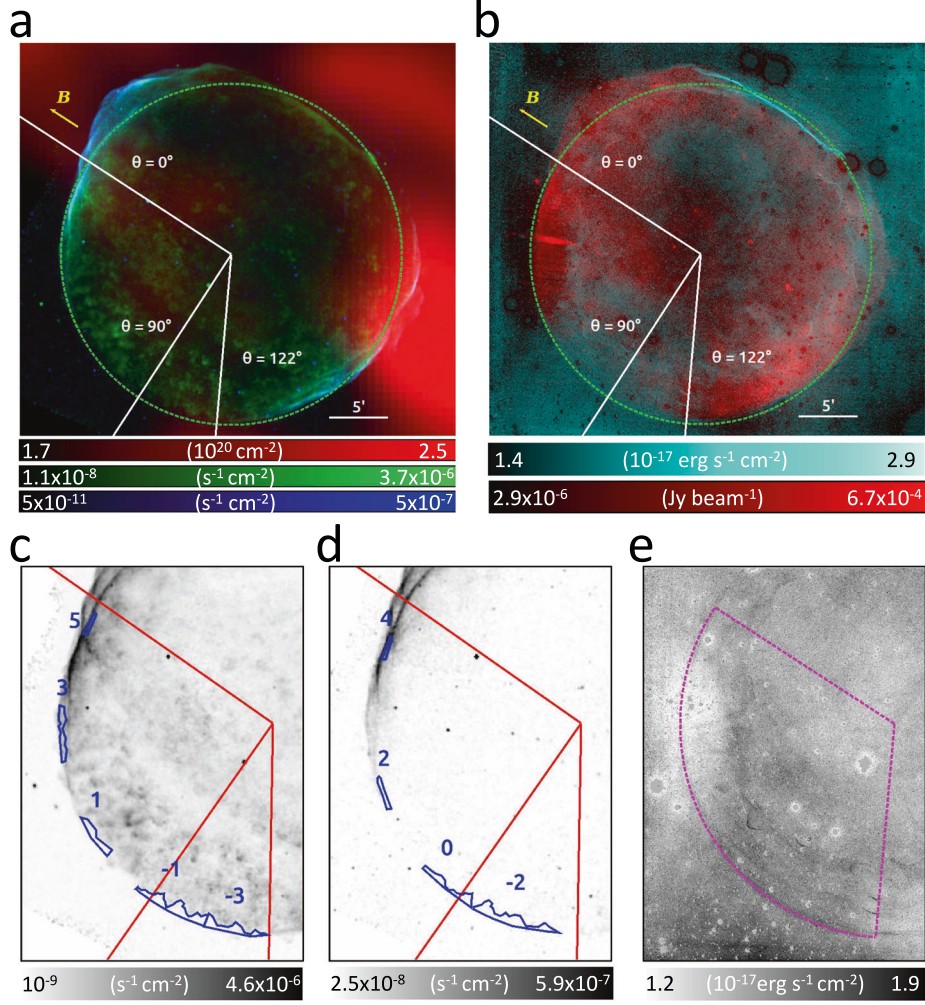

**Fig. 1 | SN 1006 in different energy bands. a** *Chandra* flux images (photons cm$^{-2}$ s$^{-1}$) in the 0.5–1 keV (green) and 2.5–7 keV (blue) bands, and column density (in 10$^{20}$ cm$^{-2}$) of HI in the [+5.8, +10.7] km s$^{-1}$ range[30] (red). Lines mark angles ($\theta$) relative to the ambient magnetic field, **B**. The white segment indicates an angular distance of 5′. **b** radio continuum map[62] (Jy beam$^{-1}$) at 1.4 GHz (red), with Balmer H$_\alpha$ emission[31] (10$^{-17}$ erg s$^{-1}$ cm$^{-2}$, in light blue). **c, d** 0.5–1 keV and 2.5–7 keV maps, respectively, with the 9 regions selected for spectral analysis superposed in blue. **e** close-up view of the H$_\alpha$ map. The dashed circle marks the position of the forward shock. Minimum and maximum values for each panel are indicated in the corresponding color bar. All scales are linear, except for the light blue in **b**, which is in square root scale.

observations of the southeastern limb of SN 1006[29]. In this region, the X-ray emission is mainly thermal but contaminated by the contribution of shocked ejecta. Nevertheless, a spatially resolved spectral analysis showed a faint X-ray emitting component associated with the shocked ISM. The density of this component can be inferred from its emission measure (*EM*, defined as the integral of the square of the thermal particle density over the volume of the X-ray emitting plasma). The post-shock density shows hints of azimuthal modulation, with a minimum around $\theta = 90°$ (i.e., quasi-perpendicular conditions) and a rapid increase toward the nonthermal limbs, suggestive of a higher shock compressibility. This analysis was restricted to a small portion of the shell (approximately $\theta = 90° \pm 35°$), without including regions with quasi-parallel conditions, and limited by the spatial resolution of the *XMM-Newton* telescope (comparable to the distance between the shock and ejecta). It was thus not possible to isolate regions with ISM emission only, and all the spectra were contaminated by the ejecta contribution, requiring additional assumptions (namely, pressure equilibrium between ejecta and ISM) to estimate the volume occupied by the ISM and then its density.

Here, we identify shock modification in SN 1006 by studying the azimuthal profile of the post-shock density, and overcome the aforementioned limitations by combining deep X-ray observations performed with *XMM-Newton* (https://www.cosmos.esa.int/web/xmm-newton) and *Chandra* telescopes (https://chandra.harvard.edu/), to benefit from spectroscopy sensitivity and high-spatial resolution of the post-shock region.

## Results

### Spatially resolved spectral analysis

Figure 1a shows the soft (0.5–1 keV, mainly thermal emission) and hard (2.5–7 keV, nonthermal emission) X-ray maps of SN 1006, together with HI distribution in the ambient medium[30]. Panel b shows the Balmer $H_\alpha$ image[31] and the radio continuum map. The ambient magnetic-field orientation is superimposed on the maps, with angle $\theta = 0°$ assumed at the center of the northeastern limb. The position of the forward shock is indicated by its Balmer optical emission (Fig. 1b, e) in thermal limbs, and by the hard X-ray synchrotron filaments in non-thermal limbs (Fig. 1a).

The spatial distribution of the ambient atomic hydrogen around SN 1006 is quite homogeneous[30,32] and the only structures detected are localized in the western and northwestern rims (Fig. 1a). None are observed in the regions considered for our analysis, where the ambient medium appears uniform. While we cannot exclude small fluctuations in the ambient density, they are undetected with current radio data. We consider two additional pieces of evidence supporting a uniform upstream medium in this sector of the remnant. The first one is the fairly circular shape of the shock front from the southern to northeastern rim: the shock velocity (and then its radius) depends on the ambient density, and the almost constant shock radius shown in Fig. 1 indicates a uniform environment. The second piece of evidence is the faint and fairly uniform surface brightness of the $H_\alpha$ emission (whose intensity depends on the local density) in this sector of the shell (Fig. 1e). These two conditions support the scenario of a tenuous and homogeneous ambient medium. Therefore, we interpret azimuthal modulations in the post-shock density as ascribed only to variations in $r_t$.

We define nine narrow spatial regions immediately behind the forward shock for X-ray spectral studies, excluding the regions contaminated by the ejecta (see lower panels of Fig. 1). The superior spatial resolution of *Chandra* allows us to identify the outermost ejecta, whose projected position in the plane of the sky is marked by ripples of thermal emission. Abrupt variations in the X-ray surface brightness of thermal emission show the position of the contact discontinuity. In particular, the X-ray surface brightness of the outermost ejecta is more than 10 times larger than that of the background ($S_{out}$, i.e., the surface brightness outside the shell), while the X-ray surface brightness within

regions in the southeastern limb (i.e., regions −3, −2,−1, 0, +1) is only $2 - 5 \times S_{out}$. The inner border of these regions corresponds to a surface brightness contour level at $6 \times S_{out}$, thus marking the sharp separation between ejecta and ISM emission. In regions 2, 3, 4, 5, the contribution of synchrotron emission to the X-ray surface brightness dominates. Therefore, in this part of the remnant we selected very narrow regions, immediately behind the shock front, by carefully excluding visible ejecta clumps. We investigate a large portion of the shell ($\theta = 0°$–122°), including regions with quasi-parallel conditions where shock modification is expected to be at its maximum.

Figure 2a shows the X-ray spectrum extracted from region 0, revealing the shocked ambient medium at approximately $\theta = 90°$, where we do not observe prominent particle acceleration and the shock is adiabatic ($r_t = 4$). The spectrum can be modelled by an isothermal optically thin plasma in non-equilibrium of ionization (parametrized by the ionization parameter $\tau$, defined as the time integral of the density reckoned from the impact with the shock). The post-shock density of the plasma is $n = 0.164^{+0.014}_{-0.016}$ cm$^{-3}$, in good agreement with previous estimates in this part of the shell[29], (as well as the electron temperature, approximately $kT = 1.35$ keV, and $\tau = 4.8^{+0.9}_{-4.7} \times 10^8$ s cm$^{-3}$), but our values are obtained without any ad-hoc assumptions. The interstellar absorption is expected to be uniform in the portion of the shell analyzed, and we fix the absorbing column density to $N_H = 7 \times 10^{20}$ cm$^{-2}$, in agreement with radio observations[32]. We detect the shocked ISM emission in all regions, with a statistical significance >99%. Figure 2b shows that the ISM contribution is clearly visible at low energies even in the spectrum of region 5 (at approximately $\theta = 0°$) where the X-ray emission is dominated by synchrotron radiation. We found that the electron temperature in the shocked ISM is consistent with being constant over all the explored regions (though with large uncertainties), and letting it free to vary does not improve the quality of the fits. So we fixed it to the best-fit value obtained in region 0 ($kT = 1.35$ keV), where we obtain the most precise estimate (see, methods subsection X-ray data analysis). In case of shock modification, we expect the ion temperature to be lower in regions with efficient hadron acceleration. This should also reduce the electron temperature, but this effect is predicted to be quite small[33]. Moreover, this reduction may be compensated by the fact that strong magnetic turbulence in quasi-parallel regions should favor electron heating (by heat exchange with ions). We expect much larger variations of the shock compression ratio, so we trace the shock modification by focusing on the post-shock density. We estimate the density from the best fit value of the *EM* in each spectral region (by numerically computing the volume of the emitting plasma, see, methods subsection X-ray data analysis).

### Azimuthal profile of the post-shock density

Figure 3a shows the azimuthal modulation of the post-shock density of the ISM. We verified that the estimates of the density are not affected by contamination from the ejecta. Ejecta can be easily identified because of their higher surface brightness with respect to the ISM. If we extract the spectra by slightly changing the shape of the extraction regions, so as that their inner boundaries include ejecta knots, the plasma density increases artificially (and the quality of the fit decreases). For example, by moving inwards the inner boundary of region 0 (region 3), and enhancing its projected area by less than 10% (<40% for region 3), we find that the plasma density changes from $n = 0.164^{+0.014}_{-0.016}$ cm$^{-3}$ up to $n = 0.189 \pm 0.011$ cm$^{-3}$ in region 0 (and from $n = 0.21^{+0.05}_{-0.04}$ cm$^{-3}$ to $n = 0.25 \pm 0.03$ cm$^{-3}$ in region 3). Conversely, by reducing the size of the extraction regions, the plasma density stays constant, thus indicating that the medium within each region is fairly uniform (e.g., by reducing the projected area of region 0 and region 3 by about 25%, we find $n = 0.158^{+0.015}_{-0.021}$ cm$^{-3}$, in region 0, and $n = 0.22^{+0.06}_{-0.05}$ cm$^{-3}$ in region 3). We conclude that Fig. 3 traces the azimuthal density modulation of the shocked ISM.

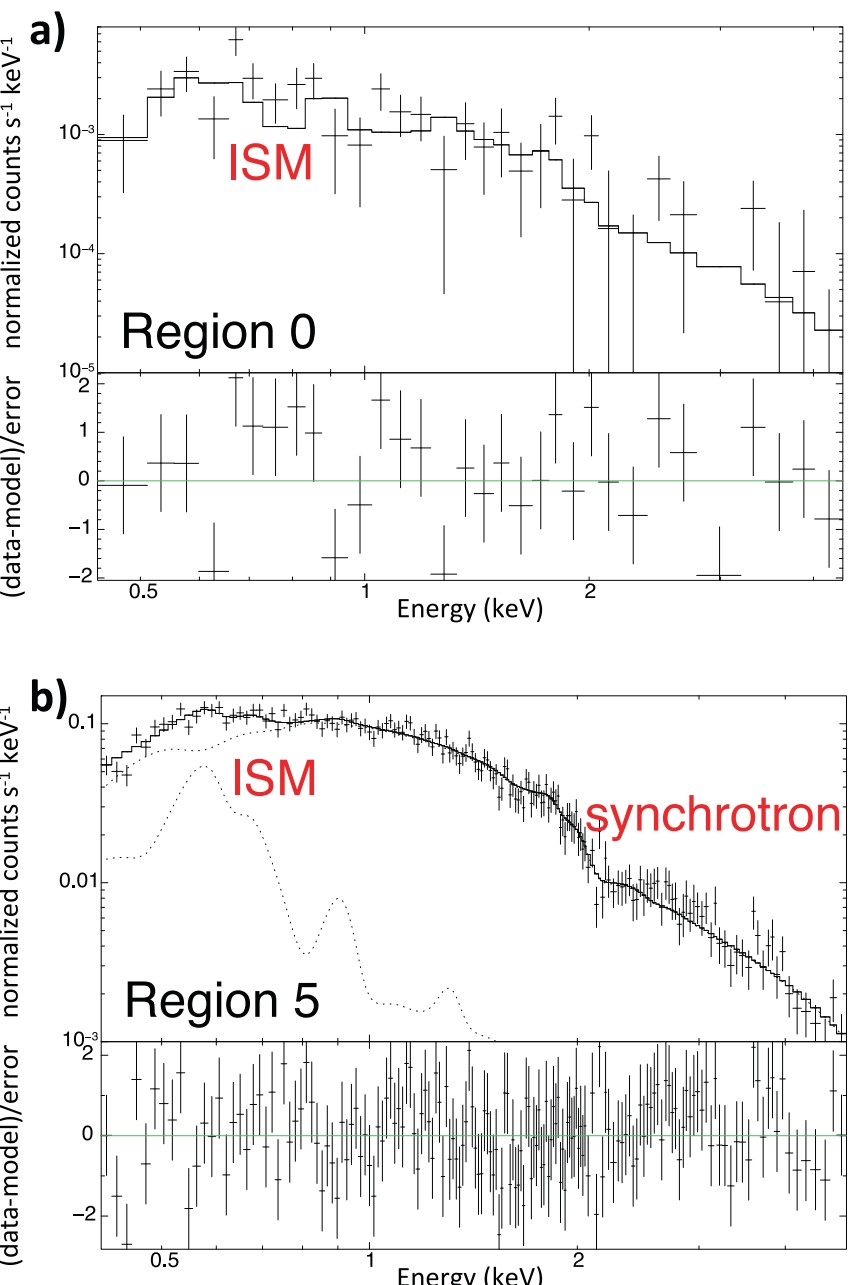

**Fig. 2 | Spatially resolved spectral analysis.** *Chandra* X-ray spectra (black crosses) of regions 0 (**a**) and 5 (**b**) of Fig. 1, with the corresponding best fit models (solid lines) and residuals (lower insets in each panel). Error bars are at the 68% confidence level. Data and models are folded through the instrumental response (ACIS-I and ACIS-S in region 0 and region 5, respectively). Thermal (ISM) and nonthermal (synchrotron) contributions are highlighted with dotted lines.

As explained above, it is thus hard to explain this density modulation as a result of inhomogeneities in the upstream medium. These inhomogeneities, if present, would affect the shock velocity ($v_s$, which is proportional to the inverse of the square root of the ambient density), inducing a $\Delta v_s$ of approximately 1200 km s$^{-1}$ between region 0 and region 5 (for a shock velocity in region 5 of[31] 5000 km s$^{-1}$). This would produce a difference in the shock radius of about 10$^{18}$ cm (corresponding to 0.5′ at 2.2 kpc[34]) between region 0 and region 5 in only 250 yr. This is at odds with observations, which show a very circular shape of the shock front (see Fig. 1e), whose radius vary less than 0.15′ between region 0 and region 5 (see, methods subsection X-ray data analysis for details). We then consider the density modulation as a tracer of azimuthal variations in the shock compression ratio. Assuming $r_t = 4$ in region 0 (i. e., at $\theta = 90°$,

where the acceleration is inefficient), Fig. 3b shows a higher compressibility in quasi-parallel conditions, where the shock compression ratio raises up to approximately $r_t = 7$.

To further constrain the observed increase of the post-shock density towards quasi-parallel conditions, we added the data from the *XMM-Newton* Large Program of observations of SN 1006 (approximately $t_{exp} = 750$ ks). We updated the previous results obtained for 8 regions in the southeastern limb[29] (from around $\theta = 55°$ to approximately $\theta = 120°$) to correct for the effects of the telescope point spread function (see, methods subsection X-ray data analysis). In addition, we extended the study to quasi-parallel regions by analyzing the *XMM-Newton* spectra including region 3 and region 4–5 (together). We adopted the same model as that adopted for *Chandra* data. The agreement between results obtained with the two telescopes is remarkable (see Fig. 3) and

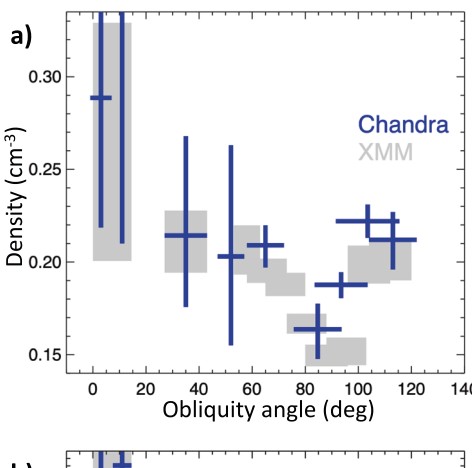

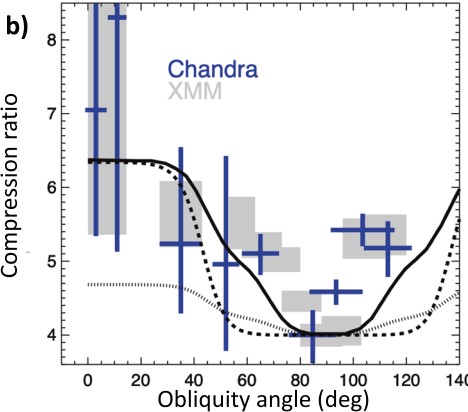

**Fig. 3 | Modulation of the shock compressibility.** Azimuthal profile of post-shock density (**a**) and compression ratio (**b**) derived from *Chandra* (blue crosses) and *XMM-Newton* (grey boxes) spectra. Errors are at the 68% confidence level. Angles are measured counterclockwise from the direction of ambient magnetic field. Compression ratios were obtained by assuming a compression ratio of 4 in *Chandra* region 0 and in *XMM-Newton* region e (see Table 1, respectively. The solid curve marks the profile expected for parallel efficiency $\xi_p = 12\%$, normalized magnetic pressure $\xi_B = 5\%$, and efficiency of CR re-acceleration $\xi_s = 6\%$, dashed curve is for $\xi_p = 18\%$ and $\xi_s = 0\%$, dotted curve is for $\xi_p = 12\%$, $\xi_s = 6\%$ and $\xi_B = 0\%$ (i. e., without including the effects of the postcursor). Source data are provided as a Source Data file.

the combination of the reliability provided by the high *Chandra* spatial resolution (which excludes contamination from ejecta) and the sensitivity of *XMM-Newton* (which provides more precise estimates) confirms the azimuthal modulation of the post-shock density.

In the *XMM-Newton* spectra, the higher post-shock density in region 4–5 with respect to region 0 is confirmed even by letting the electron temperature and interstellar absorption free to vary in the fitting process. In particular, we found that the quality of the fit of the spectrum of region 4–5 worsens significantly by imposing the plasma density to be the same as that observed in region 0, even by letting both $N_H$ and $kT$ free to vary ($\chi^2 = 182.3$ with 181 d.o.f., with $kT = 1.0^{+0.5}_{-0.3}$ keV and $N_H = 6.3 \pm 0.6 \times 10^{20}$ cm$^{-2}$, to be compared with $\chi^2 = 179.6$ with 182 d.o.f., see Table 1). Moreover, we notice that by imposing a low post-shock density in region 4-5, the best-fit value of the ionization parameter ($\tau = 1.3 \pm 0.2 \times 10^9$ s cm$^{-3}$) increases by a factor of about 2 with respect to that reported in Table 1, thus still indicating the need for a higher post-shock density in quasi-parallel regions.

## Discussions

From observations, the azimuthal profile of the post-shock density shown in Fig. 3 can be explained by a higher shock compression ratio in quasi-parallel regions than in quasi-perpendicular regions.

From theory, self-consistent kinetic simulations[22] unveiled that the injection of thermal protons into DSA is maximum at quasi-parallel regions, where efficiency can reach 10–15%, and suppressed at quasi-perpendicular shocks. Prominent shock modification ($r_t \lesssim 7$) is therefore expected only in quasi-parallel regions[20,21], while quasi-perpendicular shocks should show approximately $r_t = 4$. While injection of thermal ions is typically suppressed for[22] $\theta \gtrsim 45°$, re-acceleration of pre-existing Galactic CRs (seeds) can provide a modest acceleration efficiency (approximately 2–6%), thus playing a role in shock modification up to[35] $\theta \gtrsim 60°$, in agreement with the observed trend in Fig. 3. CR acceleration also leads to the amplification of the pre-shock magnetic field in the quasi-parallel regions, where synchrotron emission is more prominent and magnetic fields are more turbulent, as attested by radio polarization maps[25]. Also, quasi-perpendicular regions exhibit synchrotron emission in the radio but not in the X-rays, which points to the presence of GeV but not TeV electrons, again consistent with acceleration in the unperturbed Galactic magnetic field[2].

The solid curve in Fig. 3b shows the expected trend of $r_t$ vs. $\theta$, obtained by assuming a quasi-parallel acceleration efficiency $\xi_p = 12\%$, with a cut-off at approximately 45°. Such values are chosen based on self-consistent hybrid simulations[36], which attest to thermal ions being spontaneously injected into DSA for quasi-parallel shocks[37]. Always guided by simulations, which show that more oblique shocks are able to efficiently accelerate pre-existing CR seeds[38], we have also a component of reaccelerated CRs with efficiency $\xi_s = 6\%$ and cutoff at 70°. Since both accelerated and re-accelerated particles can effectively amplify the initial magnetic field, we pose the normalized magnetic pressure $\xi_B = 5\%$. We also note that magnetic-field amplification and high $r_t$ are consistent with the non-detection of X-ray emission from the precursor of SN 1006[39].

The profile shown by the solid curve in Fig. 3b is in line with the observed data-points. As a comparison, we also include the expected profiles obtained without CR re-acceleration (with $\xi_p = 18\%$, dashed curve), and without postcursor effects[20] ($\xi_p = 12\%$, $\xi_s = 6\%$, $\xi_B = 0$, dotted curve).

The theoretical curves in Fig. 3, while capturing the zero-th order azimuthal dependence of $r_t$ on $\theta$, do not account for the possible refined geometry of the magnetic field in the SN 1006 field. A comparison between radio maps and MHD simulations (not including shock modification)[24] suggests that the local magnetic field may be tilted by $\phi_B \approx 38° \pm 4°$ with respect to the line of sight, with a gradient of the field strength of the order of $\nabla |\mathbf{B}| = 1.5\mathbf{B}$ over 10 pc, roughly laying in the plane of the sky (parallel to the limbs) and pointing toward the Galactic plane. In general, a finite tilt $\phi_B$ would stack regions with different shock inclinations along each line of sight, thereby reducing the contrast between the regions with maximum and minimum $r_t$; nevertheless, since the expected $r_t$ is maximum for $\theta \lesssim 40°$ (see Fig. 3), any tilt $\phi_B \lesssim 40°$ would not induce any major modification to the curves in Fig. 3. The presence of a gradient, instead, has been shown[24] to reduce the angular distance between the two polar caps, producing a narrow minimum in the $r_t$ versus $\theta$ profile, remarkably similar to that observed (see, methods subsection Modeling the shock modification). Since the simple geometry assumed in this paper captures well the bilateral morphology of SN 1006 and its azimuthal variations, we defer the study of the corrections induced by more elaborate field geometries to a forthcoming publication.

Finally, we consider the effects that the shock modification should induce on the spectrum of the accelerated particles. In the context of the classical theory of non-linear DSA[40,41], a shock compression ratio larger than 4 leads to CR spectra harder than $E^{-2}$, while radio observations suggest for SN 1006 a radio spectral index of 0.6[42], corresponding to a CR spectrum $\propto E^{-2.2}$. Nevertheless, when the postcursor physics is taken into account in the calculation of the CR spectral index[21], for the parameters chosen in the paper for the q-parallel regions ($\xi_{tot} = \xi_p + \xi_s = 18\%$, $\xi_B = 5\%$), one obtains $r_t = 6.34$ and a CR

**Table 1 | Best fit values of emission measure (*EM*), ionization parameter (τ), cutoff energy of the synchrotron emission (*E$_{cut}$*) and post-shock density (*n$_{ISM}$*) derived from *Chandra* and *XMM-Newton* spectra extracted from the regions shown in Fig. 1, with the corresponding values of $\chi^2$ and degrees of freedom (*d.o.f.*)**

| Chandra | | | | | | |
|---|---|---|---|---|---|---|
| Region | Volume ($10^{55}$ cm$^3$) | EM ($10^{54}$ cm$^{-3}$) | τ ($10^8$ s cm$^{-3}$) | $E_{cut}$ ($10^3$ eV) | $n_{ISM}$ (cm$^{-3}$) | $\chi^2$/d.o.f. |
| 0 | 4.76 | $1.3^{+0.2}_{-0.2}$ | $4.8^{+0.9}_{-0.7}$ | | $0.164^{+0.014}_{-0.016}$ | 56.47/42 |
| +1 | 4.26 | $1.9^{+0.2}_{-0.2}$ | $6.2^{+0.6}_{-0.5}$ | | $0.209^{+0.011}_{-0.012}$ | 122.38/77 |
| +2 | 1.84 | $0.8^{+0.5}_{-0.3}$ | $7.9^{+4}_{-1.9}$ | $0.076^{+0.008}_{-0.007}$ | $0.20^{+0.06}_{-0.05}$ | 207.42/130 |
| +3 | 3.49 | $1.6^{+0.9}_{-0.5}$ | $7^{+3}_{-2}$ | $0.170^{+0.007}_{-0.006}$ | $0.21^{+0.05}_{-0.04}$ | 178.73/162 |
| +4 | 1.37 | $1.61^{+1.9}_{-1.0}$ | $5.4^{+6}_{-2.0}$ | $0.309^{+0.019}_{-0.017}$ | $0.34^{+0.17}_{-0.13}$ | 107.8/91 |
| +5 | 1.37 | $0.8^{+0.7}_{-0.4}$ | $7^{+5}_{-2}$ | $0.27^{+0.02}_{-0.02}$ | $0.29^{+0.10}_{-0.07}$ | 124.22/92 |
| −1 | 9.52 | $3.3^{+0.2}_{-0.2}$ | $4.9^{+0.4}_{-0.3}$ | | $0.188^{+0.007}_{-0.007}$ | 137.83/94 |
| −2 | 11.3 | $5.6^{+0.4}_{-0.4}$ | $4.7^{+0.4}_{-0.3}$ | | $0.222^{+0.009}_{-0.009}$ | 71.75/41 |
| −3 | 6.09 | $2.7^{+0.4}_{-0.4}$ | $5.7^{+0.8}_{-0.6}$ | | $0.212^{+0.015}_{-0.016}$ | 47.42/40 |
| XMM-Newton | | | | | | |
| Region | Volume ($10^{55}$ cm$^3$) | EM ($10^{54}$ cm$^{-3}$) | τ ($10^8$ s cm$^{-3}$) | $E_{cut}$ ($10^3$ eV) | $n_{ISM}$ (cm$^{-3}$) | $\chi^2$/d.o.f. |
| +3 | 3.43 | $1.26^{+0.26}_{-0.15}$ | 11.5 ± 2.0 | $0.138^{+0.004}_{-0.005}$ | $0.206^{+0.020}_{-0.011}$ | 168.0/176 |
| +4−5 | 2.65 | $1.5^{+0.9}_{-0.6}$ | $6.7^{+3}_{-1.9}$ | 0.272 ± 0.007 | $0.26^{+0.07}_{-0.06}$ | 179.6/182 |

Errors are at the 68% confidence level (the values of temperature and absorbing column density are fixed to $kT = 1.35$ keV, and $N_H = 7 \times 10^{20}$ cm$^{-2}$, respectively). Source data are provided as a Source Data file.

**Table 2 | List of observations analyzed in this work**

| Chandra | | | | |
|---|---|---|---|---|
| Obs ID | Instrument | Exp (ks) | PI name | link |
| 13737 | ACIS - S | 87.89 | Winkler | https://cxc.cfa.harvard.edu/cdaftp/byobsid/7/13737/ |
| 13738 | ACIS - I | 73.47 | Winkler | https://cxc.cfa.harvard.edu/cdaftp/byobsid/8/13738/ |
| 13739 | ACIS - I | 100.07 | Winkler | https://cxc.cfa.harvard.edu/cdaftp/byobsid/9/13739/ |
| 13740 | ACIS - I | 50.41 | Winkler | https://cxc.cfa.harvard.edu/cdaftp/byobsid/0/13740/ |
| 13741 | ACIS - I | 98.48 | Winkler | https://cxc.cfa.harvard.edu/cdaftp/byobsid/1/13741/ |
| 13742 | ACIS - I | 79.04 | Winkler | https://cxc.cfa.harvard.edu/cdaftp/byobsid/2/13742/ |
| 13743 | ACIS - I | 92.56 | Winkler | https://cxc.cfa.harvard.edu/cdaftp/byobsid/3/13743/ |
| 14423 | ACIS - I | 25.02 | Winkler | https://cxc.cfa.harvard.edu/cdaftp/byobsid/3/14423/ |
| 14424 | ACIS - I | 25.39 | Winkler | https://cxc.cfa.harvard.edu/cdaftp/byobsid/4/14424/ |
| 14435 | ACIS - I | 38.32 | Winkler | https://cxc.cfa.harvard.edu/cdaftp/byobsid/5/14435/ |
| 9107 | ACIS - S | 68.87 | Petre | https://cxc.cfa.harvard.edu/cdaftp/byobsid/7/9107/ |
| XMM-Newton | | | | |
| Obs ID | Instrument | Exp (ks) | PI name | link |
| 0555630201 | EPIC | 109.719 | Decourchelle | http://nxsa.esac.esa.int/nxsa-web/#obsid=0555630201 |

Source data are provided as a Source Data file.

spectrum $\propto E^{-2.19}$, in remarkable agreement with the observed radio index. In conclusion, our findings show an azimuthal modulation of the post-shock density in SN 1006, which is consistent with a substantial deviation of the shock compression ratio from the value of $r_t = 4$ (expected for strong shocks) in regions of prominent particle acceleration, where electrons are accelerated to TeV energies. The inferred values of compression ratios and CR slopes are compatible with those expected in CR-modified shocks when the effects of the postcursor are also accounted for[20,21]. Moreover, the azimuthal variation of $r_t$ (Fig. 3) attests to the prominence of parallel acceleration and to the important role played by the re-acceleration of pre-existing Galactic CRs for oblique shocks.

## Methods
### X-ray data analysis
We analyzed *Chandra* observations 13737, 13738, 13739, 13740, 13741, 13742, 13743, 14423, 14424, 14435 (PI F. Winkler) performed between

April and June 2012, with a total exposure time of 669.85 ks, and observation 9107 (PI R. Petre) performed on June 2008 for a total exposure time of 68.87 ks (see Table 2). All observations were reprocessed with CIAO 4.12[43] and CALDB 4.9.0 (https://heasarc.gsfc.nasa.gov/docs/heasarc/caldb/caldb_intro.html).

Mosaic images of SN 1006 were obtained by combining the different pointings with the CIAO task merge_obs (https://cxc.cfa.harvard.edu/ciao/ahelp/merge_obs.html). In particular, we produced vignetting-corrected mosaic images of the flux (in counts s$^{-1}$ cm$^{-2}$) in the 0.5 − 1 keV band (shown in green in Fig. 1) and in the 2.5−7 keV band (light blue in Fig. 1).

The contact discontinuity in SN 1006 is very close to the forward shock[26,27]. To measure the ISM post-shock density, we then extract X-ray spectra by selecting narrow regions between the contact discontinuity and the shock front. Regions selected for spatially resolved spectral analysis are shown in Fig. 1. By assuming $\theta = 0°$ at the center of the northeastern radio limb, the azimuthal range explored is

$\theta = 0° - 122°$. In this azimuthal range, the spherical shape of the shock front, combined with the extremely faint and uniform HI emission, clearly point toward a uniform ambient environment. We do not consider regions with negative values of $\theta$ because of the lack of spherical shape in the remnant therein, combined with the superposition of several shock fronts (which make it difficult to correctly estimate the volume of the X-ray emitting plasma). We do not consider regions with $\theta > 122°$ because it is not possible to select regions not contaminated by the ejecta emission, given that several ejecta knots reaching the shock front (and even protruding beyond it) can be observed in the soft X-ray image (Fig. 1c) for approximately $\theta = 122°–150°$. Beyond approximately $\theta = 150°$ the shell loses its spherical shape and interacts with an atomic cloud[30,44] (Fig. 1a).

Spectra, together with the corresponding Auxiliary Response File, ARF, and Redistribution Matrix File, RMF, were extracted via the CIAO tool `specextract` (https://cxc.cfa.harvard.edu/ciao/ahelp/specextract.html). Background spectra were extracted from regions selected out of the remnant, without point-like sources and, when possible, in the same chip as the source regions. We verified that the results of our spectral analysis are unaffected by the selection of other background regions. Spectra were rebinned by adopting the "optimal binning" procedure[45]. As a cross-check, we also rebinned the spectra so as to get at least 25 counts per spectral bin, obtaining the same results, though with slightly larger error bars. Spectral analysis was performed with XSPEC version12.10.1f[46] in the 0.5–5 keV band, by adopting $\chi^2$ statistics. Spectra extracted from the same region of the sky in different observations were fitted simultaneously. We found out that all our results do not change significantly by modeling the spectrum of the background, instead of subtracting it, and by using Cash statistics instead of $\chi^2$-minimization in the fitting process.

Thermal emission from the shocked ISM was described by an isothermal plasma in non equilibrium of ionization with a single ionization parameter, $\tau$ (model NEI in XSPEC, https://heasarc.gsfc.nasa.gov/xanadu/xspec/manual/node195.html, based on the database AtomDB version 3.09, see http://www.atomdb.org/Webguide/webguide.php). Though we adopt a state-of-the-art spectral model, we acknowledge that there may be limitations in the description of the emission stemming from an under-ionized plasma with a very low ionization parameter, such as the one studied here. However, we expect this effect to introduce almost the same biases (if any) in all regions, and not be responsible for the density profile shown in Fig. 3. We found that the electron temperature in the shocked ISM is consistent with being constant over all the explored regions and fixed it to the best-fit value obtained in region 0 ($kT = 1.35$ keV), where we get the most precise estimate (error bars approximately 0.4 keV at the 68% confidence level). This value is in remarkable agreement with that measured in a similar region with *XMM-Newton*[29]. We included the effects of interstellar absorption by adopting the model TBABS (https://heasarc.gsfc.nasa.gov/xanadu/xspec/manual/node268.html) within XSPEC. The interstellar absorption is expected to be uniform in the portion of the shell analyzed and we fixed the absorbing column density to $N_H = 7 \times 10^{20}$ cm$^{-2}$, in agreement with radio observations[32]. We performed the F-test in all regions, finding that the quality of the spectral fittings does not improve significantly by letting $kT$, or $N_H$ free to vary. The ISM emission measure and ionization parameter, $\tau$, were left free to vary in the fitting procedure.

We verified that this model provides an accurate description of spectra extracted from regions in the thermal southeastern limb (namely regions 0, −1, −2, −3, +1) and an additional nonthermal component does not improve significantly the quality of the fits, its normalization being consistent with 0 at less than the 99% confidence level. However, in regions +2, +3, +4, +5 there is a significant synchrotron emission. We then added a synchrotron component when fitting the spectra from these regions and modeled the synchrotron emission by considering the electron spectrum in the loss-dominated case[47], since this model is particularly well suited for SN 1006[48] (our results and conclusions do not change by adopting an exponentially cut-off power-law distribution of electrons (XSPEC/SRCUT model, https://heasarc.gsfc.nasa.gov/xanadu/xspec/manual/node228.html) to describe synchrotron emission, as done in previous works[27,29]). Normalization and break energy of the synchrotron emission were left free to vary in the fittings. The normalization of the thermal component is significantly larger than 0 at the 99% confidence level in all regions.

Table 1 shows the best fit results for all the regions, with errors at the 68% confidence level. We derive the average plasma density, $\overline{n}$, in each spectral region from the corresponding best-fit value of the emission measure of the plasma ($EM = \int n^2 dV = \overline{n^2}V$, where $n$ is the plasma density and $V$ is its volume). The volume is calculated with the following method (see Supplementary Software 1 for further details). We project the regions shown in Fig. 1 on a uniform grid with pixel size $0.2'' \times 0.2''$ ($0.2''$ correspond to about $6.3 \times 10^{15}$ cm at a distance of 2.2 kpc[34]). For each pixel, we calculate the corresponding depth as the length of the chord along the line of sight intercepted by the sphere that maps the shock front, and compute the volume accordingly, We then sum over all the pixels within a given region. The radius of the sphere marking the shock front slightly depends on the region considered, ranging from $R_{min} = 14.4'$ in region +5 to $R_{max} = 14.55'$ in region 0, but we use the same center for all the regions (namely, $\alpha = 15^h : 02^m : 55.74^s$, $\delta = -41° : 56' : 56.603''$). We verified the precision and reliability of our method by considering more regular regions, like those adopted in previous works[29], where the volume can be calculated analytically. We found differences $< 0.4\%$ between the numerical and analytical values. The volumes of the emitting plasma in the regions adopted for spectral analysis are listed in Table 1 and were used to derive $\overline{n}$ from $EM$. We verified that our results and conclusions do not change by adopting the PSHOCK model within XSPEC to model the thermal emission. The PSHOCK model assumes a linear distribution of the ionization parameter versus emission measure[49], ranging from zero (at the shock front) up to a maximum value ($\tau^{max}$ which is a free parameter in the fit), instead of a single, "mean", ionization parameter as the NEI model. The best-fit ISM density obtained in region 0 and region 5 with the PSHOCK model is $n_0^P = 0.163^{+0.014}_{-0.017}$ cm$^{-3}$ and $n_5^P = 0.32^{+0.11}_{-0.08}$ cm$^{-3}$, respectively (to be compared with $n_0 = 0.164^{+0.014}_{-0.016}$ cm$^{-3}$ and $n_5 = 0.29^{+0.10}_{-0.07}$ cm$^{-3}$ obtained with the NEI model). As expected, the maximum ionization parameter is approximately a factor of 2 higher than the mean $\tau$ obtained with the NEI model ($\tau_0^{max} = 8.9^{+2.1}_{-1.5} \times 10^8$ s cm$^{-3}$ and $\tau_5^{max} = 1.2^{+1.1}_{-0.4} \times 10^9$ s cm$^{-3}$, to be compared with $\tau_0 = 4.8^{+0.9}_{-0.7} \times 10^8$ s cm$^{-3}$ and $\tau_5 = 7^{+5}_{-2} \times 10^8$ s cm$^{-3}$).

Table 1 shows the best-fit values of the ionization parameter $\tau = \int_{t_s}^{t_f} n dt = \overline{n}\overline{\Delta t}$ (where $\overline{n}$ is the time-averaged plasma density, and $\overline{\Delta t}$ is the mean time elapsed since the shock impact within the region) in all regions. Figure 4 shows the confidence contours of the ISM density (as derived from the $EM$) and $\tau$, in region 0 and region 5. Since both $EM$ and $\tau$ depend on the plasma density, it is possible to estimate $\overline{\Delta t}$. The figure includes isochrones in the $(n, \tau)$ space, showing that we obtain very reasonable estimates of the mean time elapsed after the shock impact ($\overline{\Delta t} = 1 - 2 \times 10^2$ yr).

The radial size of the extraction regions changes from case to case, and so does their inner boundary, which is closer to the shock front in some regions (e.g., region 3, 4, 5) than in others (e.g., region −1 and −2). Therefore, $\overline{\Delta t}$ is not strictly the same for all regions, and the ionization parameter does not depend only on the plasma density (we expect lower $\overline{\Delta t}$ in the narrow regions around the northeastern polar cap). However, we find that, especially for regions with similar radial size, higher values of the post-shock density are associated with higher values of $\tau$, as shown in Table 1 and Fig. 5. The azimuthal profile of the ionization parameter shown in Fig. 5 is then consistent with the density profile shown in Fig. 3.

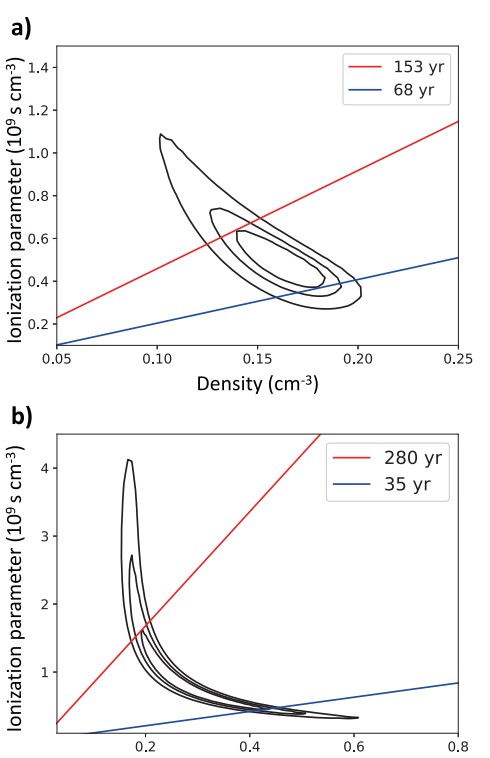

**Fig. 4 | Ionization parameter and post-shock density.** 68%, 90%, and 99% confidence contour levels of the ISM density and ionization parameter derived from the *Chandra* spectra of region 0 (**a**) and region 5 (**b**). Red and blue lines mark isochrones after the interaction with the shock front.

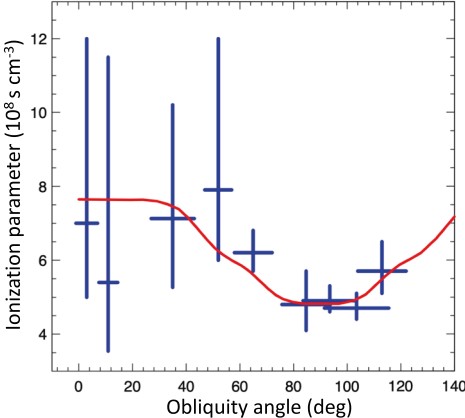

**Fig. 5 | Modulation of the ionization parameter.** Azimuthal profile of the ISM ionization parameter derived from *Chandra* spectra (blue crosses) (see Table 1). Errors are at the 68% confidence level. Angles are measured counterclockwise from the direction of ambient magnetic field. The red curve marks the profile expected for parallel efficiency ($\xi_p = 12\%$, with $\xi_B = 5\%$ and $\xi_s = 6\%$), assuming the same $\overline{\Delta T}$ for all regions. Source data are provided as a Source Data file.

In the framework of the *XMM-Newton* Large Program of observations of SN 1006 (PI A. Decourchelle), we analyze the EPIC observation 0555630201 (see Table 2). *XMM-Newton* EPIC data were processed with the Science Analysis System software, V18.0.0 (see the Users Guide to the XMM-Newton Science Analysis System", Issue 17.0, 2022 https://xmm-tools.cosmos.esa.int/external/xmm_user_support/documentation/sas_usg/USG/). Event files were filtered for soft protons contamination by adopting the ESPFILT task (https://xmm-tools.cosmos.esa.int/external/sas/current/doc/espfilt/espfilt.html), thus obtaining a screened exposure time of 89 ks, 94 ks and 51 ks for MOS1, MOS2[50] and pn[51] data, respectively. We selected events with PATTERN≤12 for the MOS cameras, PATTERN≤4 for the pn camera, and FLAG = 0 for both.

We extracted EPIC spectra from region 3 and the union of region 4 and region 5 (hereafter region 4−5, extraction regions are shown in Fig. 1). Spectra were rebinned by adopting the optimal binning procedure and spectral analysis was performed with XSPEC in the 0.5−5 keV band by adopting the model described above for the analysis of *Chandra* spectra. MOS and pn spectra were fitted simultaneously. We point out that *Chandra* and *XMM-Newton* spectra were fitted independently.

Best fit values are shown in Table 1, with errors at the 68% confidence level. Regions selected for spectral analysis are located at the rim of the shell and part of the ISM X-ray emission is spread outside the outer border of the regions, because of the relatively large point spread function of the telescope mirrors (corresponding to about 6″ full width half maximum). We quantified this effect by assuming that the ISM emission is uniformly distributed in each spectral region and found that approximately 7% of the ISM X-ray emission leaks out of each region. We address this issue by correcting the measured plasma emission measure accordingly. This has a small effect on the density estimate, considering that the density is proportional to the square root of the emission measure. However, we applied this correction to derive the density estimates shown in Table 1 and in Fig. 3 as well as to revise the previous values obtained in the southeastern limb[29] and also shown in Table 3 and Fig. 3.

## Modeling the shock modification

Efficient acceleration of CRs has always been associated with an increase in the shock compressibility[16,41,52] as due to the softer equation of state of relativistic CRs, whose adiabatic index is 4/3 (rather than 5/3) and the escape of particles from upstream, which effectively makes the shock behave as partially radiative[53,54]. In this case, though, CR spectra would become significantly harder than $E^{-2}$ above a few GeV, at odds with $\gamma$ − ray observations of individual SNRs[55,56].

Unprecedentedly-long hybrid simulations of non-relativistic shocks have recently revealed an effect that was not accounted for in the classical DSA theory, namely that the CR-amplified magnetic turbulence may have a sizable speed with respect to the shocked plasma, resulting in a postcursor, i.e., a region behind the shock where both CRs and magnetic fields drift away from the shock faster than the fluid itself[20,21]. The postcursor-induced shock modification has two main implications: on one hand, it acts as a sink of energy, which leads to an enhanced compression, and on the other hand it advects CRs away from the shock at a faster rate, which leads to steeper spectra[57]. The relevance of the postcursor is controlled by the post-shock Alfvén velocity[20] relative to the downstream fluid velocity, and can therefore be inferred from observations in which shock velocity and downstream density and magnetic field are constrained; simple estimates for both radio SNe and historical SNRs return a remarkable agreement between observations and theory[21,58].

It has been shown[20] that it is possible to calculate the shock compression ratio given the post-shock pressures in CRs and magnetic fields ($\xi_c$ and $\xi_B$), normalized to the upstream bulk pressure. We then consider the contribution of CRs injected from the thermal pool[20,22,59] and re-accelerated seeds[35], which both are expected to produce magnetic turbulence via the Bell instability for strong shocks[35,60]. The dependence on the shock obliquity $\theta$ is modelled after hybrid simulations and modulated with

$$\xi(\theta) \equiv \frac{\xi_i}{2}\left[2 + \tanh\left(\frac{\bar{\theta}_i - \theta}{\Delta\theta}\right) - \tanh\left(\frac{\pi - \bar{\theta}_i - \theta}{\Delta\theta}\right)\right]. \tag{1}$$

**Table 3 | Updated values of density of the shocked interstellar medium from previous _XMM-Newton_ data analysis[29]**

| Azimuthal opening angle (°) | Region name[29] | Density (cm⁻³) |
|---|---|---|
| 53–63 | _a_ | $0.206^{+0.2}_{-0.13}$ |
| 58–73 | _b_ | $0.197^{+0.05}_{-0.08}$ |
| 65–80 | _c_ | $0.189^{+0.05}_{-0.07}$ |
| 73–88 | _d_ | $0.169^{+0.03}_{-0.08}$ |
| 80–96 | _e_ | $0.150^{+0.05}_{-0.06}$ |
| 88–103 | _f_ | $0.152^{+0.07}_{-0.08}$ |
| 96–112 | _g_ | $0.199^{+0.09}_{-0.11}$ |
| 104–120 | _h_ | $0.201^{+0.09}_{-0.11}$ |

Errors are at 68% confidence level. Angles are measured counterclockwise from the direction of ambient magnetic field. Source data are provided as a Source Data file.

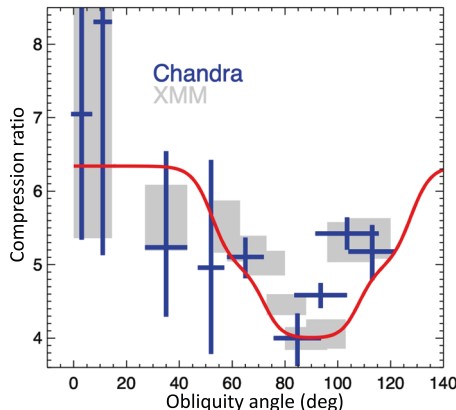

**Fig. 6 | Modulation of the shock compressibility in a non-uniform magnetic field.** Same as Fig. 3, with the solid curve marking the profile expected for $\xi_p = 12\%$, $\xi_B = 5\%$ and $\xi_s = 6\%$, but including a gradient of the magnetic field strength, lying in the plane of the sky at $\theta = 90°$. Source data are provided as a Source Data file.

We consider $i = [p, s, B]$, corresponding to the pressure in CR protons injected from the thermal pool, reaccelerated seeds, and B fields, respectively; with such a prescription, each pressure is maximum at $\xi(0) = \xi(\pi)$ and drops over an interval $\Delta\theta = 20°$ centered at $\bar{\theta}_i = [45°, 70°, 70°]$, respectively. The solid line in Fig. 3 shows the prediction for efficiencies $\xi_p = 12\%$, $\xi_s = 6\%$, and $\xi_B = 5\%$; all these values are not the result of a best fitting, but rather motivated by simulations[22,35,60] and successfully applied to the study of individual SNRs[11,58].

Note that, with the current parametrization, the total efficiency at parallel regions is $\xi_p + \xi_s \approx 18\%$, which is reasonable when acceleration of He nuclei is added on top of protons[61].

We also explored a different configuration of the ambient magnetic field, by including the effects of a gradient of the field strength (as suggested by the slantness of the radio limbs[24]) on the shock obliquity. By adopting the formalism described above, with $\xi_p = 12\%$, $\xi_s = 6\%$, and $\xi_B = 5\%$, we obtain the profile shown in Fig. 6.

## Data availability

The _Chandra_ and _XMM-Newton_ data analyzed in this paper are publicly available in the Chandra Data Archive and _XMM-Newton_ Science Archive, respectively. Table 2 provides the direct link to each observation. Datasets generated during the current study are available from the corresponding author on reasonable request. Source data are provided with this paper.

## Code availability

_Chandra_ data were processed by adopting the CIAO software package, available at https://cxc.cfa.harvard.edu/ciao/. _XMM-Newton_ data were analyzed by using the SAS package, available at https://www.cosmos.esa.int/web/xmm-newton/download-and-install-sas. XSPEC, the software adopted for X-ray spectral analysis, is available at https://heasarc.gsfc.nasa.gov/xanadu/xspec/. The Python code that we developed to calculate the volumes is provided as Supplementary Software.

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

## Acknowledgements

We thank F. Winkler for kindly providing us with the $H_\alpha$ map of SN 1006. We thank P. Plucinsky for helpful suggestions on the Chandra data analysis. M.M., S.O., F.B., E.G., and G.P. acknowledge financial contribution by the PRIN INAF 2019 grant and the INAF mainstream program. D.C. is supported by NASA grant 80NSSC20K1273 and NSF grants AST-1909778, AST-2009326 and PHY-2010240. J.V.'s and E.G.'s work on this paper has received funding from the European Union's Horizon 2020 research and innovation programme under grant agreement No 101004131 (SHARP). A.D. acknowledges support by the Centre National d'Etudes Spatiales (CNES). M.M. and R.G. acknowledge support by the INAF Mini-Grant "X-raying shock modification in supernova remnants". The scientific results reported in this article are based in part on data obtained from the Chandra Data Archive. This research has made use of

software provided by the Chandra X-ray Center (CXC) in the application package CIAO. Results are based in part on observations obtained with XMM-Newton, an ESA science mission with instruments and contributions directly funded by ESA Member States and NASA.

## Author contributions

M.M. conceived and coordinated the project, led the *XMM-Newton* data analysis and wrote the manuscript. R.G. led the analysis of the *Chandra* observations. D.C. devised the theoretical modeling of shock modification. A.D., J.V., S.O. and G.P., provided insights on the analysis and on the interpretation of the results. E.G. and F.B. collaborated to the X-ray data analysis. All authors discussed the results and implications and commented on the manuscript at all stages.

## Competing interests

The authors declare no competing interests.
