## [Peer Review File · Nature Communications]

REVIEWER COMMENTS

Reviewer #1 (Remarks to the Author):

This is an excellent and important paper, and it definitely merits publication in Nature Communications. The presentation is clear and thorough, but the paper would benefit from editing for English usage just to make it read a little more smoothly. I have only two questions.

In Figure 2, the left panel shows a purely thermal spectrum, which is around a factor of two brighter at the O VII lines around 0.56 keV than at 1 keV. The right panel shows the sum of thermal and nonthermal spectra where the temperature is forced to be the same and the value of τ is slightly higher. The ISM contribution in the right panel is negligible at 1 keV, but comparable to the synchrotron emission at .56 keV. In summary, it does not look as though the ISM spectra in the left and right panels are similar though the temperatures are the same. It looks as though the values of τ are small enough O is just beginning to be ionized to O VII, so the shape of the thermal contribution is quite sensitive to τ . It might be good to tie the density to τ .

The postcursor could be extremely important to the physics of shocks in general, but is it really possible for the cosmic rays to be advected downstream over the age of the remnant without building up in the center? In the Sedov solution, the pressure is highest in the center, and if the fraction of energy going in to CRs is constant, the CR pressure would be highest in the center unless they diffuse outward, opposite to the sense assumed. I would guess that the difference in adiabatic indices between thermal and relativistic pressure would increase this effect. I can imagine that magnetic fluctuations could damp out as they travel toward the center, but the energy would be converted to heat, tending to inhibit compression in the shocked ISM.

Reviewer #2 (Remarks to the Author):

This paper presents an observational X-ray analysis of the remnant of the historical supernova SN 1006, attempting to measure the thermal gas density immediately behind the blast wave at several points around the periphery. The point is to test a prediction of nonlinear diffusive shock-acceleration theory (NLDSA) that shock compressions are larger in the presence of efficient particle

acceleration than the adiabatic value of 4 (for monatomic ideal gases), and to spatially resolve this effect to confirm that such acceleration is more efficient when the upstream magnetic field is more closely aligned with the shock velocity (quasi-parallel acceleration) than when the two vectors are closer to orthogonal (quasi-perpendicular). The required analysis is somewhat involved, but the authors assert unequivocal evidence for three conclusions: (1) that efficient shocks do increase the compression ratio above 4; (2) that the efficiency is related to the magnetic obliquity angle θ_{Bn} between the magnetic field and the shock normal, in the way predicted by simulations; and (3) that the quantitative effect confirms recent hybrid simulation calculations (discrete ions, fluid electrons) by some of the authors. The conclusions are important, but sit on a substantial pile of assumptions and approximations. I believe the paper is worth publishing, but will require the appropriate acknowledgments of uncertainties and model assumptions.

In general, it should be kept in mind that the effect the authors are attempting to measure is a relatively small one. Figure 3 contains the basic result. But if the minimum density is larger by 30% or so, the entire effect almost vanishes. That means that it is essential to scrutinize and describe all possible sources of uncertainty. The unambiguous demonstration of this effect sets a very high bar for rigor. In particular, details of X-ray spectral analysis, normally inconsequential, can affect the results. The authors include experienced observers who are fully aware of these issues.

The steps necessary to determine a post-shock density include:

1. spatial localization and spectral analysis able to distinguish shocked interstellar medium (ISM) from shocked ejected material
2. analysis tools capable of correctly describing a plasma of extremely low ionization (by the standards of astrophysical X-ray analysis)
3. the assumption that the observed quantity, an emission measure, can be used to infer a density (this requires accurate assessment of the volume being sampled by a sky-plane region, along with the assumption of uniform density, i.e., ignoring clumping or filling-factor effects)
4. Then the densities so derived are compared with one another, and the differences attributed solely to post-shock effects (i.e.,

assuming a simple, uniform upstream medium).

All these steps can be questioned. Collectively, their effect is to reduce the level of confidence one should place in the results.

1. Figure 1 shows irregular inner boundaries of regions selected for fitting, evidently based on assumptions about the "contact

discontinuity" between shocked ISM and shocked ejecta. Since those

boundaries heavily influence the inferred emitting volume and

therefore the density, we must have confidence that they are chosen

correctly. The only information on this choice we are given is in

line 101: the extent of ejecta is asserted to be marked by "ripples of thermal emission." The fluffy nature of interior thermal emission is clear, but the idea of a clean line-of-sight separation between shocked ejecta and shocked ISM is clearly a considerable

simplification. Even if that separation is as clear as claimed, the three-dimensional volume calculated from it relies on more geometrical assumptions. The inferred density varies as the inverse square root of the volume (for an observationally determined EM), but that dependence is still strong enough to add uncertainty to the density determination; if the volume is significantly overestimated, the inferred density is too low.

2. The spectrum of selected regions is assumed to be due entirely to shocked ISM, and described by a single ionization timescale

non-equilibrium ionization (NEI) model. Both assumptions can be

questioned. A small contribution from ejecta cannot be ruled out.

Shock models, which incorporate a range of ionization timescales,

might (or might not) materially affect the results. Assumptions were made about uniformity: constant electron temperature T_e , uniform column density N_H . These are all plausible, but again, when an effect of only 30% is being sought, a high level of scrutiny needs to be applied to such assumptions. How big are the differences if T_e and N_H are allowed to differ? Or are the statistics too poor to allow unambiguous fitting then? In fact, given that backgrounds appear to have been subtracted rather than fit simultaneously, chi-square statistics are, strictly speaking, invalid when applied to unbinned spectra. If backgrounds were in fact fit, this important information should be included in the Methods section. If not, it is presumed that the binning process used by the authors produced adequate numbers of photons per bin to allow the standard (but deprecated; see, e.g.,

<https://heasarc.gsfc.nasa.gov/docs/xmm/sl/stats.html>) practice of ignoring this issue. If the authors have handled this in a more rigorous way, spelling out their procedures in the Methods section will strengthen the paper; otherwise, the reliability of the results suffers.

Even disregarding issues of statistics, X-ray spectral modeling is a tricky business, and atomic data for the very low ionization

timescales found by the authors (less than $10^9 \text{ cm}^{-3} \text{ s}$) are notoriously unreliable, as a significant fraction of the emission (even for cosmic abundances) is contributed by heavier ions with lithium-like or even lower ionization states, for which atomic data are sketchy if they exist at all. These issues are well known to

modelers, but the effect these authors are attempting to document is sufficiently subtle that a high standard of care is required, and the uncertainties so introduced are important to acknowledge.

Finally, four regions require multicomponent fitting. A particular model was used for the nonthermal component. It is a reasonable

model, but still, just a model; how strong an effect does its choice have on the results? Would a simple power-law give consistent results? Or some other curved model?

3. Given a measured EM, the inferred density also depends inversely on the square root of the filling factor. If the emitting material is concentrated in clumps, the inferred density of those clumps is larger (though the total mass in the clumps is smaller). The authors evidently assumed uniform density, which is not unreasonable, but is simply another untestable assumption. If the upstream medium is similarly clumped on small scales, the whole test performed by the authors may be unreliable. Circumstantial evidence for clumping (or for other unknown complex effects) is provided by the discrepancy between densities estimated by dividing the fitted ionization timescales by the known age (producing values from 0.015 to 0.025 cm^{-3}) and those deduced from emission measures, 10 to 20 times larger. For pure hydrogen uniformly distributed, these numbers should be very similar; the discrepancy points to effects such as clumping, nonuniform evolutionary histories, contamination by ejecta, or problems with spectral models.

4. The assumption of uniform upstream medium is essential for

resolving spatial variations in the density. There is no obvious

observational evidence for the contrary, but it is also impossible to rule out relatively small variations which would impact the results. If Region 0 has a compression ratio of 4, the inferred upstream density there is 0.04 cm^{-3} . If that density were not typical, but surrounding regions had densities of 0.05 cm^{-3} instead, the effect in Figure 3 would almost disappear, being consistent with compression ratios of 4 for all regions except near 0 degrees, where there is clear evidence for projection effects. This small variation cannot be ruled out, though the authors argue that it is unlikely.

Finally, the model curves in Figure 3 are suggestive at best. The

actual magnetic obliquity angle depends also on the angle of the

ambient magnetic field with respect to the sky plane; evidently it is assumed that the ambient field is exactly in the plane of the sky. In addition, the angular cutoffs assumed seem arbitrary. I am glad the authors do not appear to be claiming that this result confirms the recent 2D hybrid simulations of some of the authors that find a "postcursor" region of enhanced turbulence behind the shock.

In all, I believe that the results are consistent with, but do not

unambiguously confirm, the authors' claim of a systematic variation of post-shock density along the rim of SN 1006. With somewhat less

confidence, this effect can be attributed to variation of shock

acceleration efficiency with magnetic obliquity angle. I recognize

the space constraints faced by the authors, but somehow the paper must acknowledge the uncertainties I have detailed.

Several sources of uncertainty I have raised could be reduced

relatively straightforwardly. The effects of a different nonthermal

model can be easily checked. The differences in inferred density

between constant- T_e results and those where T_e is allowed to vary

could be cited, in support of the assumption of constant T_e . (If

errors are too large to confirm the density variations, that to me

would be a big warning sign.) If statistics allow, shrinking some of the regions by assuming a slightly thinner shocked-ISM region could confirm absence of ejecta contamination, or show robustness against such contamination. However, other assumptions and approximations seem unavoidable.

I have a few lesser concerns:

Line 53 and throughout: the authors say "particle acceleration" in places where they clearly mean "electron acceleration to X-ray-emitting energies" or "efficient particle acceleration".

Electrons are accelerated to nonthermal energies everywhere around the periphery of SN 1006.

Line 171: Here it should be spelled out how the contact discontinuity is located.

Line 242: A bigger effect is presumably the possible leakage of ejecta emission into the regions. Has that been taken account of in any way?

Table 1: Please include the adjusted XMM values from Ref. 29. Also, the first XMM entry (region +3) has a typo in the density.

Reviewer #3 (Remarks to the Author):

The submitted paper presents a detailed analysis of X-ray emission from the shocked region in the SN 1006.

The aim of the paper is to estimate the post-shock plasma density to evaluate the shock compression ratio.

The Authors find a remarkable correlation with the shock compression ratio and the azimuthal angle with the compression ratio increasing in the place where non-thermal X-ray emission are stronger. Such a result support the scenario where efficient particle acceleration is responsible for both the magnetic field amplification (giving rise to the observed X-ray emission) and the larger compression ratio (as a result of the larger compressibility of the plasma).

The observational results are compared with a model for particle acceleration based on PIC simulations which allow to infer an acceleration efficiency around 10-20%.

Moreover the results are double checked using two set of X-ray data obtained with Chandra and XMM-Newton.

The paper is well written, clear and the results are very significant for the high energy astrophysics community.

I recommend the paper for publication.

I just have a couple of minor comments detailed below.

- line 148. Why for reaccelerating the critical angle is assumed to be 70° and acceleration efficiency only 6%? Are those numbers suggested by PIC simulations or just a working hypothesis? I am particularly confused by the difference between the critical angles of 45° assumed for acceleration and the 70° for reacceleration. Could the Authors explain such a difference?

- Appendix M3, line 273. Also the critical angle for magnetic field amplification is assumed to be 70° and I do not understand why. The magnetic field amplification should be driven by particles acceleration. Now, given that the efficiency in acceleration is larger than efficiency in reacceleration 12% vs. 6%, I would expect the critical angle for the magnetic field to be closer to 45° . Why it is not?

- I think that some comment on the non-thermal spectrum would be in appropriate. Given the effect of post-cursor, the slope of accelerated particles in the region where acceleration efficiency is the larger should be steeper than the standard result for shock acceleration, i.e. $f(E) \sim E^{-2}$. This would be in agreement by the radio spectrum of SN 1006 which has a slope ~ 0.6 , implying an electron spectrum propto $E^{-2.2}$. I was wondering whether the Authors could comment on this point, maybe specifying the slope they predict given the compression ratio estimated from Figure 3 and the Alfvén speed as estimated from the magnetic field amplification.

Authors' Response to Reviews of

The supernova remnant SN 1006 as a Galactic particle accelerator

Roberta Giuffrida^{1,2}, Marco Miceli^{*,1,2}, Damiano Caprioli³, Anne Decourchelle⁴, Jacco Vink⁵, Salvatore Orlando², Fabrizio Bocchino², Emanuele Greco^{1,2,5}, Giovanni Peres^{1,2}

¹*Università degli Studi di Palermo, Dipartimento di Fisica e Chimica E. Segrè, Piazza del Parlamento 1, 90134 Palermo, Italy*

²*INAF-Osservatorio Astronomico di Palermo, Piazza del Parlamento 1, 90134 Palermo, Italy,*

³*Department of Astronomy and Astrophysics & Enrico Fermi Institute, The University of Chicago, 5640 S Ellis Ave, Chicago, IL 60637, USA*

⁴*CEA, Département d'Astrophysique, CEA Saclay, Gif-sur-Yvette Cedex, France*

⁵*Anton Pannekoek Institute, GRAPPA, University of Amsterdam, PO Box 94249, 1090 GE Amsterdam, The Netherlands*

Dear Editor,

We thank you and the reviewers for your time and effort in improving our paper. We have revised it accordingly and, for clarity, we now have a section (Section 3) explicitly dedicated to the discussion.

Please find below our point-by-point answers to the comments of the reviewers.

Reviewer's comments (RC) are indicated in **bold**, our answers are in red, marked by **AR** (Authors' Response).

Best regards,

The authors

1 Reviewer 1

RC: This is an excellent and important paper, and it definitely merits publication in Nature Communications. The presentation is clear and thorough, but the paper would benefit from editing for English usage just to make it read a little more smoothly.

AR: We tried to improve the readability of the manuscript. We significantly expanded the text and added a new Section (Sect.3)

RC: I have only two questions. In Figure 2, the left panel shows a purely thermal spectrum,

which is around a factor of two brighter at the O VII lines around 0.56 keV than at 1 keV. The right panel shows the sum of thermal and nonthermal spectra where the temperature is forced to be the same and the value of tau is slightly higher. The ISM contribution in the right panel is negligible at 1 keV, but comparable to the synchrotron emission at .56 keV. In summary, it does not look as though the ISM spectra in the left and right panels are similar though the temperatures are the same. It looks as though the values of tau are small enough O is just beginning to be ionized to O VII, so the shape of the thermal contribution is quite sensitive to tau.

AR: The two spectra in Fig. 2 were extracted from two different observations, performed with different detectors (namely, ACIS-I for region 0 and ACIS-S for region 5), the extraction regions being in different areas of the shell (and of the ACIS CCDs). The differences noted by the referee mainly arise from the fact that each spectrum is folded through the corresponding instrumental response. We attach here (upper panels of RFig. 1) the *unfolded* spectra of region 0 and region 5, obtained by multiplying the data points times (unfolded model)/(folded model) (Lower panels of RFig. 1 show just the best fit model of thermal emission in the two regions). This representation provides a more accurate description of thermal emission and shows that the two thermal spectra are quite similar (the flux at ~ 1 keV being about two orders of magnitude lower than that at ~ 0.56 keV, in both cases). We understand that the actual (folded) data shown in Figure 2 can be misleading, so we point out in the revised manuscript (caption of Fig. 2) that they are folded through the instrumental response. We prefer to show actual data in Figure 2, since unfolded spectra are model-dependent.

RFig. 1: *Top panels:* Unfolded *Chandra* spectrum of region 0 (*left*) and 5 (*right*), with the corresponding best fit models and residuals. Thermal (ISM) and nonthermal (synchrotron) contributions are highlighted. *Bottom panels:* best fit model of thermal emission (NEI component) in region 0 (*left*) and 5 (*right*).

RC: It might be good to tie the density to tau.

AR: We thank the referee for this suggestion (see also our reply to Reviewer 2). We added a discussion on the ionization parameter (lines 325-350 in the revised manuscript), including two new figures (Fig. 4 and Fig. 5), in the Methods section. The radial size of our extraction regions changes from case to case, and so does the inner boundary of our regions, which is closer to the shock front in some regions than in others. Therefore, the time elapsed after the shock impact is not strictly the same for all regions, and the ionization parameter does not depend only on the plasma density. However, on average, we find larger values of the ionization parameter in regions with higher post-shock density (see Fig. 5 in the revised manuscript), thus supporting the presence of higher density plasma in quasi-parallel regions.

RC: The postcursor could be extremely important to the physics of shocks in general, but is it really possible for the cosmic rays to be advected downstream over the age of the remnant without building up in the center? In the Sedov solution, the pressure is highest in the center, and if the fraction of energy going in to CRs is constant, the CR pressure would be highest in the center unless they diffuse outward, opposite to the sense assumed. I would guess that the difference in adiabatic indices between thermal and relativistic pressure would increase this effect. I can imagine that magnetic fluctuations could damp out as they travel toward the center, but the energy would be converted to heat, tending to inhibit compression in the shocked ISM.

The referee raises a very good point about the extent of the postcursor, also in the context of

the whole SNR evolution. For the postcursor's effects on shock compression¹ and on the CR spectrum², it is sufficient that the magnetic field remains amplified on a scale larger than the diffusion length of the highest-energy CRs, $\Lambda(E_{\max})$. Nevertheless, this is naturally satisfied because, if the fields were damped, the maximum CR energy would be accordingly reduced. For typical SNR parameters, $\Lambda(E_{\max})$ is much smaller than the SNR radius, and in general the region with amplified fields should be confined between the forward shock and the contact discontinuity. Beyond the latter, CRs most likely have a large diffusion length and fill the "cavity" almost isotropically; the effect of their pressure may be important and affect the overall SNR evolution³, but not what happens locally at the shock, which is probed by the X-ray observations in the paper.

2 Reviewer 2

RC: This paper presents an observational X-ray analysis of the remnant of the historical supernova SN 1006, attempting to measure the thermal gas density immediately behind the blast wave at several points around the periphery. The point is to test a prediction of non-linear diffusive shock-acceleration theory (NLDSA) that shock compressions are larger in the presence of efficient particle acceleration than the adiabatic value of 4 (for monatomic ideal gases), and to spatially resolve this effect to confirm that such acceleration is more efficient when the upstream magnetic field is more closely aligned with the shock velocity (quasi-parallel acceleration) than when the two vectors are closer to orthogonal (quasi-perpendicular). The required analysis is somewhat involved, but the authors assert unequivocal evidence for three conclusions: (1) that efficient shocks do increase the compression ratio

above 4; (2) that the efficiency is related to the magnetic obliquity angle θ_{Bn} between the magnetic field and the shock normal, in the way predicted by simulations; and (3) that the quantitative effect confirms recent hybrid simulation calculations (discrete ions, fluid electrons) by some of the authors. The conclusions are important, but sit on a substantial pile of assumptions and approximations. I believe the paper is worth publishing, but will require the appropriate acknowledgments of uncertainties and model assumptions.

In general, it should be kept in mind that the effect the authors are attempting to measure is a relatively small one. Figure 3 contains the basic result. But if the minimum density is larger by 30% or so, the entire effect almost vanishes. That means that it is essential to scrutinize and describe all possible sources of uncertainty. The unambiguous demonstration of this effect sets a very high bar for rigor. In particular, details of X-ray spectral analysis, normally inconsequential, can affect the results. The authors include experienced observers who are fully aware of these issues. The steps necessary to determine a post-shock density include:

1. spatial localization and spectral analysis able to distinguish shocked interstellar medium (ISM) from shocked ejected material
2. analysis tools capable of correctly describing a plasma of extremely low ionization (by the standards of astrophysical X-ray analysis)
3. the assumption that the observed quantity, an emission measure, can be used to infer a density (this requires accurate assessment of the volume being sampled by a sky-plane)

region, along with the assumption of uniform density, i.e., ignoring clumping or filling-factor effects)

4. Then the densities so derived are compared with one another, and the differences attributed solely to post-shock effects (i.e., assuming a simple, uniform upstream medium).

All these steps can be questioned. Collectively, their effect is to reduce the level of confidence one should place in the results.

AR: We understand these concerns and we address all of them point-by-point below.

RC: 1. Figure 1 shows irregular inner boundaries of regions selected for fitting, evidently based on assumptions about the "contact discontinuity" between shocked ISM and shocked ejecta. Since those boundaries heavily influence the inferred emitting volume and therefore the density, we must have confidence that they are chosen correctly. The only information on this choice we are given is in line 101: the extent of ejecta is asserted to be marked by "ripples of thermal emission."

AR: We added a paragraph in the revised version of the manuscript to explain in detail how we determine the projected position of the contact discontinuity (lines 110-119). The surface brightness of the ejecta is definitely larger than that of the ISM. The inner border of the regions in the southeastern thermal limb corresponds to the contour level at a surface brightness 6 times larger than that of the background. As an example, we show region -1 and region +1 with the corresponding contour levels in the attached RFig. 2

RFig. 2: *Chandra* flux image of the southeastern limb of SN 1006 in the 0.5 – 1 keV band. Contour levels at 6 times the average background are superimposed together with regions -1 and +1.

RC: The fluffy nature of interior thermal emission is clear, but the idea of a clean line-of-sight separation between shocked ejecta and shocked ISM is clearly a considerable simplification. Even if that separation is as clear as claimed, the three-dimensional volume calculated from it relies on more geometrical assumptions. The inferred density varies as the inverse square root of the volume (for an observationally determined EM), but that dependence is still strong enough to add uncertainty to the density determination; if the volume is significantly overestimated, the inferred density is too low.

AR: We point out that there are no geometrical assumptions on the 3-D morphology of the ejecta in our calculations of the volume. The volume is calculated as the intersection of a sphere and a solid, extending along the line of sight, whose projected area (in the plane of the sky) is defined by the extraction region. By excluding from the extraction region the projection in the plane of the sky of

an ejecta knot, we remove from the corresponding volume the entire column (extending along the line of sight) including that knot. Please notice that this is a very conservative approach: we are not making any assumptions to reconstruct the 3-D distribution of the ejecta fingers; we simply remove everything that includes, in projection, a clump of ejecta. RFig 3 shows a schematic representation of how the volume is computed. We added further details on the procedure adopted for the volume calculation in the Methods section, (lines 314-319) and we attach to the revised version of the manuscript the Python script adopted.

RFig. 3: Schematic representation of the procedure adopted to calculate the volume of the X-ray emitting plasma within the extraction regions. Dark blue areas indicate the ejecta, light blue areas the shocked ISM, red areas the ISM within the extraction region. Left panel shows a representative extraction region in the plane of the sky. North (N), East (E) and line of sight (LoS) are also indicated. The panel on the right shows a slice in the E-LoS plane of the remnant and the selected region, as observed from North, after performing a 90° rotation about the axis marked by the dashed line.

RC: 2. The spectrum of selected regions is assumed to be due entirely to shocked ISM, and described by a single ionization timescale non-equilibrium ionization (NEI) model. Both assumptions can be questioned. A small contribution from ejecta cannot be ruled out. Shock models, which incorporate a range of ionization timescales, might (or might not) materially affect the results.

AR: We include a more detailed discussion on these points in the revised version of the manuscript. Regions were selected so as to remove ejecta contamination. In particular, besides the careful region selection explained above, we performed further checks by enlarging/shrinking our extraction regions. In the revised manuscript we show that by increasing the size of the regions (thus including parts of ejecta), the ejecta contamination affects the best-fit parameters, by artificially enhancing the plasma density. Conversely, by reducing the size of the regions, the plasma density stays constant, thus confirming that our regions are not affected by ejecta contamination. We comment on this point in the revised version of the manuscript (lines 147-158).

We also verified that our results do not change by adopting spectral models including a continuous distribution of ionization timescales. In particular, we fitted the spectrum of region 0 and region 5 by adopting the PSHOCK model. This model assumes a plane-parallel shock with a linear distribution of the ionization parameter vs plasma emission measure (which may not be an accurate approximation in our irregular regions, as shown in Rfig 3). The estimates of the ISM density are unaffected, while, as expected, the upper values of the ionization parameter obtained with PSHOCK are a factor ~ 2 higher than those obtained with the NEI model. We added a paragraph

to discuss this point in the Methods section (lines 325-335).

RC Assumptions were made about uniformity: constant electron temperature T_e , uniform column density N_H . These are all plausible, but again, when an effect of only 30% is being sought, a high level of scrutiny needs to be applied to such assumptions. How big are the differences if T_e and N_H are allowed to differ? Or are the statistics too poor to allow unambiguous fitting then?

AR: We performed the F-test in all regions (lines 295-297 in the revised manuscript), finding that the quality of the spectral fittings does not improve significantly by adding one more free parameter (e.g., by letting kT_e , or N_H free to vary). Reducing the degrees of freedom beyond necessity generates entanglements between the free parameters and increases the number of local minima in the χ^2 space. We then adopt the Occam's principle of parsimony. However, we agree with the referee that it is necessary to assess whether assumptions of uniform electron temperature and absorbing column density affect our conclusions. Therefore, prompted by the referee's comment, we verified that the density contrast that we measure between region 0 and region 5 is confirmed, even by letting the electron temperature and interstellar absorption free to vary. To this end, we analyzed the XMM-Newton spectrum of region 4-5 by imposing the plasma density therein to be the same as that observed in region 0, but letting both N_H and kT_e as free parameters. We obtained a description of the data points significantly worse than that obtained with high plasma density, thus confirming that the ISM density in region 4-5 is significantly higher than that in region 0. Moreover, we notice that by imposing low post shock density in region 4-5, the best-fit value of τ

increases by a factor of ~ 2 ($\tau \sim 1.3 \pm 0.2 \times 10^9 \text{ s cm}^{-3}$), thus supporting the scenario of a denser shocked ISM in quasi-parallel regions. We added a detailed comment on this point in Sect 2 (lines 181-190).

RC: In fact, given that backgrounds appear to have been subtracted rather than fit simultaneously, chi-square statistics are, strictly speaking, invalid when applied to unbinned spectra. If backgrounds were in fact fit, this important information should be included in the Methods section. If not, it is presumed that the binning process used by the authors produced adequate numbers of photons per bin to allow the standard (but deprecated; see, e.g., <https://heasarc.gsfc.nasa.gov/docs/xmm/sl/stats.html>) practice of ignoring this issue. If the authors have handled this in a more rigorous way, spelling out their procedures in the Methods section will strengthen the paper; otherwise, the reliability of the results suffers.

AR: Chandra and XMM-Newton spectra were rebinned by adopting the optimal binning procedure (Kaastra & Bleeker 2016) as explained in the Methods section. We point out that the X-ray background in XMM-Newton EPIC spectra is very different from that in Chandra ACIS observations (cfr. <https://www.cosmos.esa.int/web/xmm-newton/epic-background-components> and <https://cxc.harvard.edu/contrib/maxim/bg/>), but we obtain consistent results between the two cameras with our analysis. As a cross-check, we also rebinned the spectra so as to get at least 25 counts per spectral bin, obtaining the same results, though with slightly larger error bars. We added a comment on this point in the revised manuscript (lines 278-279).

RC: Even disregarding issues of statistics, X-ray spectral modeling is a tricky business, and

atomic data for the very low ionization timescales found by the authors (less than 10^9 cm^{-3} s) are notoriously unreliable, as a significant fraction of the emission (even for cosmic abundances) is contributed by heavier ions with lithium-like or even lower ionization states, for which atomic data are sketchy if they exist at all. These issues are well known to modelers, but the effect these authors are attempting to document is sufficiently subtle that a high standard of care is required, and the uncertainties so introduced are important to acknowledge.

AR: We agree with the referee and added a comment on this point (lines 284-288). We cannot exclude that there may be some biases in the best fit values we obtained, because of the limitations in the (state-of-the-art) spectral models adopted. However, we expect that this effect (if present) should be almost the same for all regions, and not be responsible for the density profile shown in Fig. 3.

RC: **Finally, four regions require multicomponent fitting. A particular model was used for the nonthermal component. It is a reasonable model, but still, just a model; how strong an effect does its choice have on the results? Would a simple power-law give consistent results? Or some other curved model?**

AR: The model we adopted to describe synchrotron radiation is based on the exact asymptotic solutions of the kinetic equation describing the electron energy distribution in the loss-limited case (Zirakashvili & Aharonian, 2007, hereafter loss-lim model). We found that the loss-lim model provides the best description of the synchrotron X-ray emission in SN 1006 (Miceli et al. 2013). We follow the referee's suggestion and adopt the SRCUT model (which instead assumes an exponen-

tially cut off power-law distribution of electrons) to describe the synchrotron emission in regions +2, +3, +4, +5. The quality of the fits is still good, but slightly worse than that obtained with the loss-lim model ($\Delta\chi^2 = 1 - 10$, depending on the region). In any case, the thermal component is not affected by the choice of the model adopted to describe synchrotron X-ray emission, as shown in the attached RTab. 1. We added a comment on this point in the Methods section (lines 305-307).

Region	τ	E_{cut}	n_{ISM}	χ^2/dof
(Loss Lim.)	(10^8 s cm^{-3})	(keV)	(cm^{-3})	
+2	$7.9^{+4}_{-1.9}$	$0.076^{+0.008}_{-0.007}$	$0.20^{+0.06}_{-0.05}$	207.42/130
+3	7^{+3}_{-2}	$0.170^{+0.007}_{-0.006}$	$0.21^{+0.05}_{-0.04}$	178.73/162
+4	$5.4^{+6}_{-2.0}$	$0.309^{+0.019}_{-0.017}$	$0.34^{+0.17}_{-0.13}$	107.8/91
+5	7^{+5}_{-2}	$0.27^{+0.02}_{-0.02}$	$0.29^{+0.10}_{-0.07}$	124.22/92
Region	τ	E_{break}	n_{ISM}	χ^2/dof
(SRCUT)	(10^8 s cm^{-3})	(keV)	(cm^{-3})	
+2	7^{+3}_{-2}	$0.047^{+0.007}_{-0.005}$	$0.20^{+0.05}_{-0.04}$	211.44/130
+3	7^{+3}_{-2}	$0.137^{+0.007}_{-0.006}$	$0.22^{+0.06}_{-0.04}$	188.75/162
+4	$5.0^{+5}_{-1.5}$	$0.283^{+0.017}_{-0.016}$	$0.36^{+0.16}_{-0.13}$	108.6/91
+5	7^{+3}_{-2}	$0.240^{+0.019}_{-0.015}$	$0.30^{+0.10}_{-0.07}$	124.52/92

RTab. 1: Best fit parameters (error bars at 90% confidence level) obtained by adopting the loss limited model (upper) and the SRCUT model (lower) to describe nonthermal X-ray emission.

RC: 3. Given a measured EM, the inferred density also depends inversely on the square root of the filling factor. If the emitting material is concentrated in clumps, the inferred density of those clumps is larger (though the total mass in the clumps is smaller). The authors evi-

dently assumed uniform density, which is not unreasonable, but is simply another untestable assumption. If the upstream medium is similarly clumped on small scales, the whole test performed by the authors may be unreliable. Circumstantial evidence for clumping (or for other unknown complex effects) is provided by the discrepancy between densities estimated by dividing the fitted ionization timescales by the known age (producing values from 0.015 to 0.025 cm^{-3}) and those deduced from emission measures, 10 to 20 times larger. For pure hydrogen uniformly distributed, these numbers should be very similar; the discrepancy points to effects such as clumping, nonuniform evolutionary histories, contamination by ejecta, or problems with spectral models.

AR: If the emission is concentrated in a few small (with respect to the size of the region) clumps, the best fit value of the density should be highly sensitive to the size of the extraction regions. As explained above, and in the revised manuscript (lines 154-157) this is not the case, and by reducing the size of the extraction region we recover the same values of the ISM density.

Fig. 4 of the revised manuscript also shows that there are no discrepancies between density estimates and ionization timescales, since we obtain very reasonable values for the time elapsed after the shock impact. We note that the NEI model provides a mean value of the ionization parameter ($\tau \sim \tau_{max}/2$, where τ_{max} is the maximum ionization parameter in the region). Moreover, to avoid ejecta contamination, we exclude from our regions part of the plasma close to the contact discontinuity (i.e. the ISM with the highest ionization parameter, see right panel of RFig. 3). Therefore, we expect to recover a time elapsed after the shock impact definitely lower than half of the age of

SN 1006, which is indeed what we get.

Finally, at the observed spatial resolution (all multi-wavelength data included), there is no hint of small scale inhomogeneities along the shock, which would trace the presence of a clumpy medium. In particular, a clumpy ISM should provide a knotty X-ray emission (the plasma emissivity being proportional to the square of the particle density) that we do not observe in the X-ray map. While the soft X-ray emission is extremely knotty in the ejecta (thus reflecting their clumpy structure), the surface brightness of the shocked ISM (between contact discontinuity and forward shock) appears fairly uniform.

RC: The assumption of uniform upstream medium is essential for resolving spatial variations in the density. There is no obvious observational evidence for the contrary, but it is also impossible to rule out relatively small variations which would impact the results. If Region 0 has a compression ratio of 4, the inferred upstream density there is 0.04 cm^{-3} . If that density were not typical, but surrounding regions had densities of 0.05 cm^{-3} instead, the effect in Figure 3 would almost disappear, being consistent with compression ratios of 4 for all regions except near 0 degrees, where there is clear evidence for projection effects. This small variation cannot be ruled out, though the authors argue that it is unlikely.

AR: Though there may be projection effects in region 5 (but we expect that these effects, if any, might reduce, not enhance, the density estimate therein), this is not the case for region 4, which was chosen in a “clean” part of the shock front. Therefore, to explain the bilateral profile as a result of pre-shock inhomogeneities would require a very ad-hoc configuration of the ambient medium. Our

argument supporting the uniform ambient medium is based on i) radio HI observations (showing no visible inhomogeneities in the upstream medium), ii) uniform (albeit very faint) H_α emission (whose intensity depends on the local density), and iii) fairly circular shape of the shock front (whose velocity depends on the upstream density) in the sector of the shell considered. We agree that tiny (currently non-detectable) inhomogeneities in the upstream density distribution cannot be excluded, and added some words of cautions in lines 97-98. However, we point out that these complex inhomogeneities in the pre-shock density would induce local variations in the shock velocity and radius, which would be much higher than those observed, as we show in the revised manuscript (lines 159-167). We then conclude that interpreting the post-shock density profile as a result of variations in the shock compression factor provides the most physically sound (and the simplest) explanation.

RC: Finally, the model curves in Figure 3 are suggestive at best. The actual magnetic obliquity angle depends also on the angle of the ambient magnetic field with respect to the sky plane; evidently it is assumed that the ambient field is exactly in the plane of the sky. In addition, the angular cutoffs assumed seem arbitrary. I am glad the authors do not appear to be claiming that this result confirms the recent 2D hybrid simulations of some of the authors that find a "postcursor" region of enhanced turbulence behind the shock.

AR: As noticed by the referee, we are not trying to overinterpret the data, and Fig. 3 just aims at showing that the density profile, which clearly hints at an azimuthal modulation, can be explained by state-of-the-art models of shock modification.

In terms of modeling, Fig. 3 in the paper conveys the message that standard shock modification (magenta curve) alone cannot account for a compression ratio larger than about 4.7 even for a $\sim 20\%$ CR efficiency. As the referee points out, the plot *suggests* that the presence of a postcursor, as outlined in recent hybrid simulations, may close the gap between theory and data.

In this respect, in the revised manuscript we stress that the chosen angular cutoffs are not arbitrary (see also our reply to Reviewer 3): the contribution of DSA driven by spontaneously-injected thermal ions cuts off at about 45° (figure 3 of Caprioli & Spitkovsky 2014a)⁴ and the contribution of reaccelerated CR seeds cuts off at about $60 - 70^\circ$ (figure 6 of Caprioli et al. 2018)⁵. We added a comment on this point in the revised manuscript (lines 206-213).

The actual 3-D configuration of the ambient magnetic field in SN 1006 is still unknown. Therefore, as noticed by the referee, we consider the simplest case in Fig. 3, by assuming an aspect angle of 0° . It has been shown⁶ that the pronounced slantness of the synchrotron radio limbs of SN 1006 can be naturally explained by a gradient of the field strength, $\nabla|\vec{B}| = 1.5B$ over 10 pc, running parallel to the limbs (in the plane of the sky, pointing toward the Galactic plane) and forming an angle consistent with zero with the plane of the sky. We point out that, in this framework, the angular distance between the two polar caps is lower than 180° and this produces a narrow minimum in the profile, in nice agreement with the data points (see Fig. 6 and lines 404-407 in the revised manuscript).

If the aspect angle of the magnetic field with respect to the line of sight is $\phi_B \neq 0^\circ$ the projected position of the polar caps might not lie at the limb of the shell, and this may reduce the contrast

between the maximum and minimum values of r_t . A detailed comparison between radio maps and MHD simulations (not including effects of shock modification) indicates⁶ $\phi_B = 38^\circ \pm 4^\circ$. For this inclination with respect to the ambient magnetic field, we still expect $r_t \sim 6.3$ (see Fig. 3 in the manuscript). Therefore, the maximum and minimum shock compression ratios are not expected to be significantly affected by this issue. We expect, however, some variations in the shape of the r_t vs. θ profile. Accounting for these fine effects requires a detailed deprojection study, which is beyond the scope of this work. Moreover, it would be first necessary to revise the methodology adopted by Bocchino et al. 2011, including the effects of shock modification.

From this exercise we conclude that our results are not strongly affected by small/ moderate tilts of the local magnetic field with respect to the assumed symmetry. In addition, adding a gradient would move the second maximum of the compression ratio closer to 90° , hence improving the agreement between theory and data. We added a detailed discussion about all these points in Sect. 3 (lines 219-233) of the revised manuscript and included a new figure (Fig. 6) in the Methods section.

RC: In all, I believe that the results are consistent with, but do not unambiguously confirm, the authors' claim of a systematic variation of post-shock density along the rim of SN 1006. With somewhat less confidence, this effect can be attributed to variation of shock acceleration efficiency with magnetic obliquity angle. I recognize the space constraints faced by the authors, but somehow the paper must acknowledge the uncertainties I have detailed. Several sources of uncertainty I have raised could be reduced relatively straightforwardly. The

effects of a different nothermal model can be easily checked. The differences in inferred density between constant- T_e results and those where T_e is allowed to vary could be cited, in support of the assumption of constant T_e . (If errors are too large to confirm the density variations, that to me would be a big warning sign.) If statistics allow, shrinking some of the regions by assuming a slightly thinner shocked-ISM region could confirm absence of ejecta contamination, or show robustness against such contamination. However, other assumptions and approximations seem unavoidable.

AR: We addressed possible sources of uncertainty in our replies above. We also added more details in the revised version of the manuscript to show that our conclusions are unaffected by letting the electron temperature free to vary, and by shrinking the size of the extraction regions. We excluded ejecta contamination and clumpiness in the interstellar medium. We also show that pre-existing inhomogeneities in the ambient medium are unlikely. Based on this, in the manuscript we present our conclusions, i.e. that an azimuthal modulation of the shock compression ratio provides the best explanation for the observed density profile (lines 192-193).

Finally, we note that variation of the acceleration efficiency with angle alone would not be sufficient to explain the observed large values of r_t . In the standard non-linear DSA theory, without the dynamical modifications induced by a postcursor, a 20% efficiency would return $r_t = 4.3$ only; also, a large r_t would produce hard CR spectra, at odds, e.g., with the radio spectral index of 0.6 (as for the spectral index in SN 1006, please see also our reply to Reviewer 3.)

RC: I have a few lesser concerns:

RC: Line 53 and throughout: the authors say "particle acceleration" in places where they clearly mean "electron acceleration to X-ray-emitting energies" or "efficient particle acceleration". Electrons are accelerated to nonthermal energies everywhere around the periphery of SN 1006.

AR:Right, we corrected this issue everywhere in the text

RC: Line 171: Here it should be spelled out how the contact discontinuity is located.

AR: As explained above (see also RFig. 2), we included a detailed discussion on this issue in the revised manuscript (lines 110-119).

RC: Line 242: A bigger effect is presumably the possible leakage of ejecta emission into the regions. Has that been taken account of in any way?

AR: Yes, spectral modeling of "thermal" regions analyzed in Miceli et al. (2012) also include an ejecta component and ISM densities were derived by assuming pressure equilibrium between ejecta and ISM (lines 80-82 in the manuscript, see Miceli et al. 2012 for details). Region 3 and region 4-5 are much narrower than thermal regions and located further away from the contact discontinuity, so we assumed no leakage from ejecta therein. The agreement with *Chandra* results confirms the validity of our assumption.

RC: Table 1: Please include the adjusted XMM values from Ref. 29. Also, the first XMM entry (region +3) has a typo in the density.

AR: We included the adjusted XMM values in Table 2 in the revised manuscript. We apologize for the typos in the Table. We corrected the typo in region 3, together with a few other typos in Table 1 (all corrections are highlighted in blue). All our conclusions are unaffected.

3 Reviewer 3

RC: The submitted paper presents a detailed analysis of X-ray emission from the shocked region in the SN 1006. The aim of the paper is to estimate the post-shock plasma density to evaluate the shock compression ratio. The Authors find a remarkable correlation with the shock compression ratio and the azimuthal angle with the compression ratio increasing in the place where non-thermal X-ray emission are stronger. Such a result support the scenario where efficient particle acceleration is responsible for both the magnetic field amplification (giving rise to the observed X-ray emission) and the larger compression ratio (as a result of the larger compressibility of the plasma).

The observational results are compared with a model for particle acceleration based on PIC simulations which allow to infer an acceleration efficiency around 10-20%.

Moreover the results are double checked using two set of X-ray data obtained with Chandra and XMM-Newton.

The paper is well written, clear and the results are very significant for the high energy astrophysics community. I recommend the paper for publication.

I just have a couple of minor comments detailed below.

RC: - line 148. Why for reaccelerating the critical angle is assumed to be 70° and acceleration efficiency only 6%? Are those numbers suggested by PIC simulations or just a working hypothesis? I am particularly confused by the difference between the critical angles of 45° assumed for acceleration and the 70° for reacceleration. Could the Authors explain such a difference?

AR: We added a sentence in the revised manuscript to clarify the choice of critical angles (lines 206-213). In particular, as shown in figure 3 of Caprioli & Spitkovsky (2014a)⁴, the contribution of DSA driven by spontaneously-injected thermal ions is expected to cut off at about 45° , while the contribution of reaccelerated CR seeds cuts off at about 60–70 deg (figure 6 of Caprioli et al. 2018⁵).

RC: - Appendix M3, line 273. Also the critical angle for magnetic field amplification is assumed to be 70° and I do not understand why. The magnetic field amplification should be driven by particles acceleration. Now, given that the efficiency in acceleration is larger than efficiency in reacceleration 12% vs. 6%, I would expect the critical angle for the magnetic field to be closer to 45° . Why it is not?

AR: As shown in Caprioli et al. (2018)⁵, also reaccelerated particles can drive magnetic field amplification. In particular, that paper estimates that Galactic CRs should be sufficient for driving the Bell instability, with a growth time of about 1 yr. The saturation of the Bell instability is

typically controlled by the advection time on the precursor length-scale, which is the same for accelerated and reaccelerated particles (e.g., Bell et al. 2013, Blasi et al. 2015); in other words, the total amount of magnetic field amplification is not expected to vary a lot with the shock inclination, provided that reacceleration is active ($\theta \lesssim 70^\circ$). While we agree with the referee that ξ_B may be expected to decrease above 45° , the extent of the non-thermal X-ray emission to relatively large θ suggests —on a pure phenomenological basis— that the field is indeed amplified even for $\theta > 45^\circ$. Indications for an amplified magnetic field at relatively large θ were indeed found in SN 1006⁷.

RC: - I think that some comment on the non-thermal spectrum would be in appropriate. Given the effect of post-cursor, the slope of accelerated particles in the region where acceleration efficiency is the larger should be steeper than the standard result for shock acceleration, i.e. $f(E) \sim E^{-2}$. This would be in agreement by the radio spectrum of SN 1006 which has a slope ~ 0.6 , implying an electron spectrum propto $E^{-2.2}$. I was wandering whether the Authors could comment on this point, maybe specifying the slope they predict given the compression ratio estimated from Figure 3 and the Alfvén speed as estimated from the magnetic field amplification.

AR: We really thank the referee for the opportunity to perform a check that we overlooked, since we did not focus on the spectrum of the accelerated particles in this manuscript. It turns out that with the parameters chosen in the paper for the q-parallel regions ($\xi_{tot} = \xi_p + \xi_s = 0.18$, $\xi_B = 0.06$), when applying the formalism of Caprioli et al. (2020) one obtains $r_t = 6.59$ and a CR slope of $q = 2.2057$, in remarkable agreement with the observed radio index. We added this

important piece of information in the conclusions (lines 234-241).

1. Haggerty, C. C. & Caprioli, D. Kinetic Simulations of Cosmic-Ray-Modified Shocks I: Hydrodynamics. *arXiv e-prints* arXiv:2008.12308 (2020). 2008.12308.
2. Caprioli, D., Haggerty, C. C. & Blasi, P. Kinetic Simulations of Cosmic-Ray-Modified Shocks II: Particle Spectra. *arXiv e-prints* arXiv:2009.00007 (2020). 2009.00007.
3. Diesing, R. & Caprioli, D. Effect of cosmic rays on the evolution and momentum deposition of supernova remnants. *Physical Review Letters* **121**, 091101 (2018). URL <http://adsabs.harvard.edu/abs/2018PhRvL.121i1101D>. 1804.09731.
4. Caprioli, D. & Spitkovsky, A. Simulations of Ion Acceleration at Non-relativistic Shocks: I. Acceleration Efficiency. *ApJ* **783**, 91 (2014). 1310.2943.
5. Caprioli, D., Zhang, H. & Spitkovsky, A. Diffusive shock re-acceleration. *JPP* (2018). URL <http://adsabs.harvard.edu/abs/2018arXiv180101510C>. 1801.01510.
6. Bocchino, F., Orlando, S., Miceli, M. & Petruk, O. Constraints on the local interstellar magnetic field from non-thermal emission of SN1006. *A&A* **531**, A129 (2011). 1105.2689.
7. Ressler, S. M. *et al.* Magnetic Field Amplification in the Thin X-Ray Rims of SN 1006. *ApJ* **790**, 85 (2014). 1406.3630.

Reviewers' comments:

Reviewer #1 (Remarks to the Author):

The response to the referee answers my questions, and I am happy to see that the values of $\tau = n_e t$ support the other estimates of shock compression.

This work is a qualitative advance over reference 29 in that the much higher spatial resolution of Chandra is very important for removing doubts about contamination of the ISM emission by ejecta emission. It is also unique in presenting support for the theoretical prediction of postcursor particle acceleration.

The paper is ready for publication.

Reviewer #2 (Remarks to the Author):

The authors have done a good job buttressing the likelihood of their interpretation. I appreciate the extra effort that has gone into answering some of my questions, for instance checking a different nonthermal model for the two-component fits, and trying shock models as well as single-component NEI models. (At the same time, some of the rebuttals are subject to question, most prominently those involving comparisons of regions 0 and 5, since region 5 is clearly seen in projection against another piece of rim, with unknown consequences for the analysis.) And for the alternate model for the nonthermal component: the XSPEC SRCUT model requires, or predicts, a radio flux and spectral index from the region being fitted. What values were used, or deduced?

In most cases, however, there is no way to rule out some sources of uncertainty, and my fundamental problem with the paper remains.

Unlikely but possible circumstances cannot be excluded, so the most

that can be claimed for the results is support for, but not proof of, the authors' hypothesis. The observational evidence for uniform upstream density cited by the authors is insufficient to exclude the kind of variations that could alter the conclusions. (For instance, the faint H alpha filament along the SW rim seems to vary substantially in brightness, at least in the published versions I could find.)

The argument from constancy of shock velocity can't eliminate the possibility of 30%-level small-scale fluctuations that in 100 years (the kind of shock age found by the authors when dividing their fitted ionization timescale by their fitted density) would not produce detectable deviations. The separation of ejecta and shocked ISM cannot be done perfectly (so the question becomes: how much contamination of one by the other would negate the results?)

Most problematic, though, is the basic spectral analysis, where the fitting procedure used by the authors (background subtraction and

chi-squared statistics) is known to misrepresent uncertainties. Much of the argument relies on statistics: which model best represents the data? If the varying-temperature fits are no better in some sense than the single-temperature ones, which should we believe? Absent any external evidence, fixing the electron temperature to the same value for all regions clearly underestimates uncertainties. (Occam's razor is a fine guide to relative likelihood of hypotheses, but it is hardly a demonstration. Sometimes more complicated theories are correct.) What is a reliable confidence interval for the determination of emission measure? If error bars have been underestimated by even a relatively small factor, the conclusions weaken substantially. Again, the procedure used by the authors is widespread, but that doesn't make it optimal, and in a case where a very strong assertion is being put forward, all due care should be taken in the analysis.

Thus, while this paper does a more careful job than the earlier effort in this direction (Miceli et al. 2012), it does not in my opinion represent a qualitative advance in certainty. Such an advance may simply not be possible at present, given the data available for this source.

An additional element has been added in the revision: a discussion of the radio spectrum. Higher compressibility (due to particle escape or the smaller adiabatic index of relativistic particles) leads to a uniformly steeper spectrum only for test-particle acceleration. In the versions of non-linear diffusive shock acceleration (NLDSA) most often found in the literature, a concave curvature (flattening to higher particle energies) is deduced when diffusion lengths increase with particle energy as is normally assumed (this argument goes back to Eichler 1979), producing typically steeper radio spectra for SNRs as well as radio supernovae. In fact, extrapolation of radio spectral indices of young SNRs to the X-ray significantly underpredicts the observed synchrotron X-ray emission, requiring curvature. The authors

offer an alternative explanation, but the main paper text misrepresents the predictions of conventional NLDSA, which are quite consistent with the observed steeper radio spectra of SNRs.

Reviewer #3 (Remarks to the Author):

The Authors have addressed my concerns in a satisfactory way, hence I can recommend the manuscript for publication in Nature communications.

Point-by-point response to the reviewers' comments on The supernova remnant SN 1006 as a Galactic particle accelerator

Roberta Giuffrida^{1,2}, Marco Miceli^{*,1,2}, Damiano Caprioli³, Anne Decourchelle⁴, Jacco Vink⁵,
Salvatore Orlando², Fabrizio Bocchino², Emanuele Greco^{1,2,5}, Giovanni Peres^{1,2}

¹*Università degli Studi di Palermo, Dipartimento di Fisica e Chimica E. Segrè, Piazza del Parlamento 1, 90134 Palermo, Italy*

²*INAF-Osservatorio Astronomico di Palermo, Piazza del Parlamento 1, 90134 Palermo, Italy,*

³*Department of Astronomy and Astrophysics & Enrico Fermi Institute, The University of Chicago, 5640 S Ellis Ave, Chicago, IL 60637, USA*

⁴*CEA, Département d'Astrophysique, CEA Saclay, Gif-sur-Yvette Cedex, France*

⁵*Anton Pannekoek Institute, GRAPPA, University of Amsterdam, PO Box 94249, 1090 GE Amsterdam, The Netherlands*

Please find below our point-by-point answers to the comments of Reviewer 2 (R2). Reviewer's comments are indicated in red and enumerated with roman numerals. Our answers are in black, marked by AR (Authors' Response).

I. The authors have done a good job buttressing the likelihood of their interpretation. I appreciate the extra effort that has gone into answering some of my questions, for instance checking a different nonthermal model for the two-component fits, and trying shock models as well as single-component NEI models. (At the same time, some of the rebuttals are subject to question, most prominently those involving comparisons of regions 0 and 5, since region 5 is clearly seen in projection against another piece of rim, with unknown consequences for the analysis.) And for the alternate model for the nonthermal component: the XSPEC SRCUT model requires, or predicts, a radio flux and spectral index from the region being fitted. What values were used, or deduced?

AR: In our previous reply, we pointed out that we expect that putative projection effects in region 5, if any, might reduce, not enhance, the density estimate therein (the shock breakout in nearby regions being indicative of a local reduction of the ambient density). We also notice that our results do not depend on region 5 alone, but on the coherent azimuthal trend obtained for the 9 Chandra and 2 XMM-Newton regions in Table 1 and for the 8 additional XMM-Newton regions in Table 2. As for the SRCUT model, we stress here that we performed the analysis with a more accurate model, based on the exact asymptotic solutions of the kinetic equation describing the electron energy distribution in the loss-limited case (Zirakashvili & Aharonian, 2007). By following the

suggestion of R2, however, we verified that all our conclusions are unaffected even by adopting a less refined, purely phenomenological model, SRCUT. In our manuscript, we explain (line 307) that, for the test suggested by R2, we follow the same procedure as Miceli et al. 2009 (flux at 1 GHz set to the value derived from the radio map by Petruk et al. 2009 by allowing it to vary within its uncertainties) and Miceli et al. 2013 (spectral index $\alpha = 0.5$, but results do not change by letting it free to vary, see their Sect. 2).

II. In most cases, however, there is no way to rule out some sources of uncertainty, and my fundamental problem with the paper remains. Unlikely but possible circumstances cannot be excluded, so the most that can be claimed for the results is support for, but not proof of, the authors' hypothesis. The observational evidence for uniform upstream density cited by the authors is insufficient to exclude the kind of variations that could alter the conclusions. (For instance, the faint H alpha filament along the SW rim seems to vary substantially in brightness, at least in the published versions I could find.)

AR: The referee invokes a peculiar axisymmetric density structure, specifically tuned to mimic the observed azimuthal variation of the compression ratio, which is neither expected by models nor supported by any observational evidence. We do not know what specific image the reviewer is referring to. We are referring to the 2010 map published in Winkler et al. 2013 (their Fig. 3, upper panel). The $H\alpha$ emission is extremely uniform in regions -2,-1,0,+1, (median and mean values are consistent with being the same), while it appears fainter in regions +2, +3, +4 and +5, where the flux is *lower* (and, therefore, the pre-shock density should be lower too). The continuum-subtracted

image is noisy in this part of the shell, so we adopt the *conservative* choice of assuming the same pre-shock density everywhere. However, if we wanted to consider the variations suggested by R2, we should conclude that the pre-shock density in quasi-parallel region (+2, +3, +4 and +5) is slightly *lower* than in quasi-perpendicular regions, so the compression ratio therein is even higher than that shown in Fig. 3 of our paper and this would *strengthen our conclusion*.

III. The argument from constancy of shock velocity can't eliminate the possibility of 30%-level small-scale fluctuations that in 100 years (the kind of shock age found by the authors when dividing their fitted ionization timescale by their fitted density) would not produce detectable deviations.

AR: We quantitatively demonstrated in lines 159-166 of our manuscript that fluctuations in the pre-shock density would produce macroscopic variations in the shock radius, which are not observed. We also point out that the time to be considered in the calculation is not 100 yr, as explained below. As stated in lines 336-342, we derive an average time $\overline{\Delta t} = 100 - 200$ yr, which is indicative of the mean "shock age". The time to be considered in the calculation for the shock proper motion is the maximum shock age, i. e., $\sim 2 \times \overline{\Delta t}$ (as explained in lines 332-333), so the correct value is 200 – 400 yr. We then conclude that the argument from constancy of shock velocity is robust.

We also point out that the best-fit value of the density in region 5 (4), is a factor of 1.75 (2) higher than that in region 0 (as derived both for Chandra and XMM data analysis.)

IV. The separation of ejecta and shocked ISM cannot be done perfectly (so the question becomes: how much contamination of one by the other would negate the results?)

AR: We are certain that our process of separation of ejecta and shocked ISM is done with a very high level of accuracy, and we do not find in the reports by the reviewers any quantitative argument against it. We carefully addressed this issue in the revised manuscript, by showing that 1) the separation between ejecta and ISM in the plane of the sky is sharp and the contact discontinuity is clearly detectable as an abrupt increase in surface brightness (lines 110-119 of the manuscript); 2) the choice of extraction regions and the algorithm adopted to calculate the volume are extremely conservative (as explained in pages 9-10 of our previous reply) and are specifically designed to avoid ejecta contamination (pages 9-10 of our previous reply, and RFig. 2 and RFig. 3 therein); 3) We performed a specific test on this issue, showing the reliability of our methodology, as clearly explained in lines 146-158 of the manuscript.

Most importantly, we remark that we clearly separate ejecta and ISM in the analysis of Chandra spectra (and XMM spectra of regions 3 and 4-5), while XMM spectra of regions *a – h* **include both ISM and ejecta emission** (see Table 2 in the manuscript for the updated values of density). In these regions, ISM and ejecta are *not* separated, their emission being modelled by considering two different thermal components (we explained this point at pag. 22 of our previous reply). The results obtained with the two independent methods, applied to data collected with two different instruments (i. excluding the ejecta thanks to the superior Chandra spatial resolution, and ii. modeling together ejecta and ISM) provide perfectly consistent results, as clearly shown in Fig. 3 of the manuscript.

V. Most problematic, though, is the basic spectral analysis, where the fitting procedure used by the

authors (background subtraction and chisq statistics) is known to misrepresent uncertainties.

AR: Our results are robust, as demonstrated below. R2 argues that our results may change by modeling the background, instead of subtracting it, and by using C-statistics, instead of χ^2 statistics. This is not the case, as shown in RTab. 1, where we compare the results obtained by following our procedure with those obtained by following the procedure suggested by the referee: all the conclusions are unaffected. We also point out that we do not use C-statistics because XMM-Newton spectra are weighted with the SAS `evigweight` task (Arnaud 2001, *Astron. & Astrophys.*, 365, 80, i.e., counts are corrected for vignetting) so, strictly speaking, their distribution is not Poissonian and C-statistics should *not* be applied.

In general, we feel that the referee skepticism is based on the fact that they consider the measured variations in density small, and they argue that the background or any other small fluctuations can affect this measure. Nevertheless, we remark that the variations in density are inferred from the variations in the X-ray flux, which scales as the square of the particle density for an optically thin plasma (for fixed temperature and ionization parameter). As the density in regions 4 (or 5) is approximately a factor 1.75 higher than in region 0, its thermal emission is approximately a factor 3 higher. This means that variations in the X-ray flux are macroscopic, and are well above both the background and statistic fluctuations. To show this point, we plot in RFig. 1 the spectrum of regions 4 and 5, obtained by adopting region 2 as a background (all spectra are extracted from the same Chandra observation, with ObsID 9107). The model (indicated by the black curve in RFig. 1) shows synchrotron radiation, while the residuals at low energy clearly demonstrate the excess

Model: source + bkg			
Region	density (c-stat)	C	d.o.f
	(cm^{-3})		
0	$0.165^{+0.010}_{-0.010}$	65.89	42
+3	$0.23^{+0.07}_{-0.05}$	182.88	163
+5	$0.27^{+0.10}_{-0.07}$	123.63	92

Model: source - bkg (paper)			
Region	density (chi-sq)	χ^2	d.o.f
	(cm^{-3})		
0	$0.164^{+0.014}_{-0.016}$	56.47	42
+3	$0.21^{+0.05}_{-0.04}$	178.73	162
+5	$0.29^{+0.10}_{-0.07}$	124.22	92

RTab. 1: Best-fit results. The first set of values was obtained by modelling the background and using C-statistic (as suggested by R2). The second set was obtained with background subtraction and χ^2 statistic (as in the manuscript). All errors are at the 68% confidence level.

of thermal emission in regions 4 (and 5) (i. e., quasi-parallel configuration) with respect to region 2 (quasi-perpendicular shock). As explained above, this excess is macroscopic and significant.

VI. Much of the argument relies on statistics: which model best represents the data?

Yes, spectral analysis is based on statistics. In particular, we adopt the simplest model which provides an accurate description of the observed spectra. We also note that our model is physically sound, and our conclusions show a solid trend that correlates the shock compression ratio with

RFig. 1: *Chandra* spectrum of region 4 (*left*) and 5 (*right*), subtracted by the spectrum of region 2 (taking into account the different extraction areas). Black curves show the synchrotron model. Residuals clearly indicate the excess of thermal emission in region 4 (5) with respect to region 2.

another observable, the local shock inclination. We used the theoretical insight gained from state-of-the-art numerical simulations to interpret this correlation as a signature of a varying acceleration efficiency. To our knowledge, this has never been either observed or modeled before. In summary, our approach allows us to obtain a coherent and complete physical picture of the system.

VII. If the varying-temperature fits are no better in some sense than the single-temperature ones, which should we believe? Absent any external evidence, fixing the electron temperature to the same value for all regions clearly underestimates uncertainties. (Occam's razor is a fine guide to relative likelihood of hypotheses, but it is hardly a demonstration. Sometimes more complicated theories are correct.) What is a reliable confidence interval for the determination of emission measure? If error bars have been underestimated by even a relatively small factor, the conclusions weaken substantially. Again, the procedure used by the authors is widespread, but that doesn't

make it optimal, and in a case where a very strong assertion is being put forward, all due care should be taken in the analysis.

AR: We already addressed this point at lines 181-190 of the manuscript, where we show that, by letting the electron temperature free to vary, our conclusions are unaffected.

Generally, we agree with the referee that sometimes Occam's razor may fail and the correct explanation may not be the simplest one, but here the next simplest explanation would require a theory that does not exist and adding many more parameters that cannot be constrained by observations.

VIII. Thus, while this paper does a more careful job than the earlier effort in this direction (Miceli et al. 2012), it does not in my opinion represent a qualitative advance in certainty. Such an advance may simply not be possible at present, given the data available for this source.

AR: We consider this work as a major leap forward with respect to Miceli et al. 2012, because: i) it provides a solid proof of the goodness of previous indications (by analyzing different data sets and adopting different procedures), ii) it significantly extends the previous results, by inspecting the shock modification process in an unprecedented wide range of shock obliquity angles, iii) it reveals for the first time the azimuthal dependence of the shock compression ratio, and iv) it provides, for the first time, a self-consistent physical picture, supported by state-of-the-art models.

IX. An additional element has been added in the revision: a discussion of the radio spectrum. Higher compressibility (due to particle escape or the smaller adiabatic index of relativistic par-

ticles) leads to a uniformly steeper spectrum only for test-particle acceleration. In the versions of non-linear diffusive shock acceleration (NLDSA) most often found in the literature, a concave curvature (flattening to higher particle energies) is deduced when diffusion lengths increase with particle energy as is normally assumed (this argument goes back to Eichler 1979), producing typically steeper radio spectra for SNRs as well as radio supernovae. In fact, extrapolation of radio spectral indices of young SNRs to the X-ray significantly underpredicts the observed synchrotron X-ray emission, requiring curvature. The authors offer an alternative explanation, but the main paper text misrepresents the predictions of conventional NLDSA, which are quite consistent with the observed steeper radio spectra of SNRs.

AR: The “conventional” NLDSA theory that reviewer 2 invokes to explain the radio spectral index was developed decades ago, as pointed out by the referee, but appears to be inconsistent with recent observations. NLDSA allows the spectrum to become steeper than E^{-2} at low energies only if it became (much) flatter than E^{-2} at high energies (e.g., Malkov & Drury 2001, Reports on Progress in Physics, Volume 64, Issue 4, pp 429). To reproduce the radio spectral slope observed in SN 1006 and other historical SNRs, efficiencies larger than 50% would be required, which would imply that at GeV-TeV energies the spectrum should be as flat as $E^{-1.5}$. This is at odds with actual γ -ray observations, which show a characteristic spectral index of 2.5-3 (e.g., Caprioli 2011, Journal of Cosmology and Astroparticle Physics, 5, 26; Aharonian et al. 2019, Nature Astronomy, Volume 3, p. 561-567). Also, recent kinetic simulations on one hand ruled out the emergence of concave spectra, and on the other hand provided evidence that moderate efficiencies ($\sim 10 - 20\%$) naturally lead to spectra in agreement with observations (Caprioli et al. 2020, Astrophys. J. 905, 2, Diesing

et al. 2021, *Astrophys. J.* 922, 1).

In general, it has been widely shown that conventional NLDSA is at odds with multiwavelength observations of radio SNe and SNRs (e.g., Caprioli 2012, *J. Cosmol. Astropart. Phys.* 7, 38; Bell et al. 2013, *Mon. Not. R. Astron. Soc.*, 431, 415; Malkov & Aharonian 2019, *Astrophys. J.*, 881, 13; Bell et al. 2019, *Mon. Not. R. Astron. Soc.* 488, 2466; etc.).

Finally, NLDSA does neither predict nor explain why there should be an azimuthal dependence of acceleration efficiency and/or magnetic field amplification, which is the main point of our paper.

REVIEWER COMMENTS

Reviewer #2 (Remarks to the Author):

I recognize and appreciate the effort the authors made to improve the statistical treatment. It should have been done throughout and included in the paper, as readers deserve to see the results of a more rigorous analysis. However, I did not mean to give the impression that addressing this point would fundamentally change my assessment: this paper represents an improvement, but not a qualitatively different outcome, from that of the earlier work. There are simply too many areas where even the best analysis of the data available must fall short of the level of certainty the authors claim. Untestable assumptions include, in addition to that of an upstream medium whose density is uniform to better than 30% or so, that the minimum post-shock density corresponds to a compression ratio of 4, and that the upstream magnetic field is uniform and the magnetic obliquity (not an observable, though constrained by the radio polarimetry) can be determined at each location. (There are other more technical assumptions involved in the spectral modeling.) While these assumptions are plausible, they cannot be confirmed directly, and no work based on them can be characterized as a "proof." The authors have shown fairly convincingly that there is a minimum in the post-shock thermal gas density at one location around the south rim of the remnant. It is plausible that this is connected to the efficiency of particle acceleration through the variation of the compression

ratio. But the compression ratio has not been measured; this

interpretation is not demanded by the data. (The authors' Figure 3 does not inspire confidence in the particular model they illustrate; the curves are hardly fits to the data. They simply show qualitative trends which the data appear to follow.) So as I set out in my first report, these are good results, worth publishing, but only after some moderation of the claims.

Point-by-point response to the reviewers' comments on The supernova remnant SN 1006 as a Galactic particle accelerator

Roberta Giuffrida^{1,2}, Marco Miceli^{*,1,2}, Damiano Caprioli³, Anne Decourchelle^{4,5}, Jacco Vink⁶,
Salvatore Orlando², Fabrizio Bocchino², Emanuele Greco^{6,7,2}, Giovanni Peres^{1,2}

¹*Università degli Studi di Palermo, Dipartimento di Fisica e Chimica E. Segrè, Piazza del Parlamento 1, 90134 Palermo, Italy*

²*INAF-Osservatorio Astronomico di Palermo, Piazza del Parlamento 1, 90134 Palermo, Italy,*

³*Department of Astronomy and Astrophysics & Enrico Fermi Institute, The University of Chicago, 5640 S Ellis Ave, Chicago, IL 60637, USA*

⁴*Université Paris-Saclay, CEA, CNRS, AIM, F-91191, Gif-sur-Yvette, France*

⁵*Université de Paris, AIM, F-91191 Gif-sur-Yvette, France*

⁶*Anton Pannekoek Institute, GRAPPA, University of Amsterdam, PO Box 94249, 1090 GE Amsterdam, The Netherlands*

⁷*GRAPPA, University of Amsterdam, Science Park 904, 1098 XH Amsterdam, The Netherlands*

Please find below our point-by-point answers to the comments of Reviewer 2 (R2). Reviewer's comments are indicated in red. Our answers are in black, marked by AR (Authors' Response).

I recognize and appreciate the effort the authors made to improve the statistical treatment. It should have been done throughout and included in the paper, as readers deserve to see the results of a more rigorous analysis.

AR: We added a sentence (lines 282-284) to explicitly state that our results are insensitive to the statistics adopted (χ^2 vs. C-stat) and to the treatment of the background (i. e., modeling the spectrum of the background, instead of subtracting it). We remark that we preferred to adopt χ^2 -minimization in the paper because XMM-Newton spectra are weighted with the SAS `evigweight` task (Arnaud 2001, *Astron. & Astrophys.*, 365, 80, i.e., counts are corrected for vignetting) so, strictly speaking, their distribution is not Poissonian and C-statistics should not be applied.

However, I did not mean to give the impression that addressing this point would fundamentally change my assessment: this paper represents an improvement, but not a qualitatively different outcome, from that of the earlier work. There are simply too many areas where even the best analysis of the data available must fall short of the level of certainty the authors claim. Untestable assumptions include, in addition to that of an upstream medium whose density is uniform to better than 30% or so, that the minimum post-shock density corresponds to a compression ratio of 4, and

that the upstream magnetic field is uniform and the magnetic obliquity (not an observable, though constrained by the radio polarimetry) can be determined at each location. (There are other more technical assumptions involved in the spectral modeling.) While these assumptions are plausible, they cannot be confirmed directly, and no work based on them can be characterized as a "proof." The authors have shown fairly convincingly that there is a minimum in the post-shock thermal gas density at one location around the south rim of the remnant. It is plausible that this is connected to the efficiency of particle acceleration through the variation of the compression ratio. But the compression ratio has not been measured; this interpretation is not demanded by the data. (The authors' Figure 3 does not inspire confidence in the particular model they illustrate; the curves are hardly fits to the data. They simply show qualitative trends which the data appear to follow.) So as I set out in my first report, these are good results, worth publishing, but only after some moderation of the claims.

AR: We followed the referee's suggestion and rephrased the manuscript as follows:

- Lines 7-9: the sentence "We provide a solid evidence for this effect, and probe its dependence on the orientation of the ambient magnetic field, by analyzing deep X-ray observations of the Galactic remnant of SN 1006 " was removed and replaced by "We identify this effect, and probe its dependence on the orientation of the ambient magnetic field, by analyzing deep X-ray observations of the Galactic remnant of SN 1006."
- Lines 9-11: the sentence "By comparing our results with state-of-the-art models, we conclude that SN 1006 is an efficient source of CRs and obtain an observational proof supporting

the quasi-parallel acceleration mechanism.” was removed and replaced by the sentence “By comparing our results with state-of-the-art models, we conclude that SN 1006 is an efficient source of CRs and obtain an observational support for the quasi-parallel acceleration mechanism”.

- Lines 105-106: the sentence “Therefore, azimuthal modulations in the post-shock density can be ascribed only to variations in r_t .” was removed, and replaced by the sentence “Therefore, we interpret azimuthal modulations in the post-shock density as ascribed only to variations in r_t ”.
- Lines 166-167: the sentence “We conclude that the density modulation can be considered as a tracer of azimuthal variations in the shock compression ratio” was removed and replaced by the sentence “We then consider the density modulation as a tracer of azimuthal variations in the shock compression ratio”
- Lines 171-172: the sentence “To further constrain the observed increase of the compression ratio towards quasi-parallel conditions” was removed and replaced by the sentence “To further constrain the observed increase of the post-shock density towards quasi-parallel conditions”
- Lines 192-193: the sentence “From observations, the azimuthal profile of the post-shock density shown in Fig. 3 is best explained by a higher shock compression ratio in quasi-parallel regions than in quasi-perpendicular regions.” was removed and replaced by the sentence “From observations, the azimuthal profile of the post-shock density shown in Fig. 3 can be explained by a higher shock compression ratio in quasi-parallel regions than in

quasi-perpendicular regions.”

- Line 215: the sentence “The profile shown in red in Fig. 3 provides a good description of the observed data-points” was removed and replaced by “The profile shown in red in Fig. 3 is in line with the observed data-points”. As for the comparison between model and observation, we also point out that in lines 218-219 we state that the model only captures the zero-th order dependence of the shock compression ratio on the obliquity angle.
- Lines 241-242: the sentence “In conclusion, our findings show that the shock compression ratio in SN 1006 deviates substantially from the value of $r_t \simeq 4$ (expected for strong shocks) in regions of prominent particle acceleration...” was removed and replaced by “In conclusion, our findings show an azimuthal modulation of the post-shock density in SN 1006, which is consistent with a substantial deviation of the shock compression ratio from the value of $r_t \simeq 4$ (expected for strong shocks) in regions of prominent particle acceleration.”

We also removed the word “clearly” in a few sentences: namely at line 104 (...~~clearly~~ support...), line 109 (...~~clearly~~ identify the outermost ejecta...), line 168 (...~~clearly~~ shows a higher compressibility in quasi-parallel conditions...), and line 264 (...~~clearly~~ point toward a uniform ambient environment)